# Implicitly Guided Design with PropEn: Match your Data to Follow the Gradient

**Nataša Tagasovska** [*]  **Vladimir Gligorijević** [*]  **Kyunghyun Cho** [‡]  **Andreas Loukas** [*]

## Abstract

Across scientific domains, generating new models or optimizing existing ones while meeting specific criteria is crucial. Traditional machine learning frameworks for guided design use a generative model and a surrogate model (discriminator), requiring large datasets. However, real-world scientific applications often have limited data and complex landscapes, making data-hungry models inefficient or impractical. We propose a new framework, *PropEn*, inspired by "matching", which enables implicit guidance without training a discriminator. By matching each sample with a similar one that has a better property value, we create a larger training dataset that inherently indicates the direction of improvement. Matching, combined with an encoder-decoder architecture, forms a domain-agnostic generative framework for property enhancement. We show that training with a matched dataset approximates the gradient of the property of interest while remaining within the data distribution, allowing efficient design optimization. Extensive evaluations in toy problems and scientific applications, such as therapeutic protein design and airfoil optimization, demonstrate PropEn's advantages over common baselines. Notably, the protein design results are validated with wet lab experiments, confirming the competitiveness and effectiveness of our approach. Our code is available at https://github.com/prescient-design/propen.

## 1 Introduction

Navigating the complex world of design is a challenge in many fields, from engineering [35] to material science [33, 40] and life sciences [6]. In life sciences, the goal may be to refine molecular structures for drug discovery [6], focusing on properties like binding affinity or stability. In engineering, optimizing the shapes of aircraft wings or windmill blades to achieve desired aerodynamic traits like lift and drag forces is crucial [35]. The common thread in these fields is the *design cycle*: experts start with an initial design and aim to improve a specific property. Guided by intuition and expertise, they make adjustments and evaluate the changes. If the property improves, they continue this iterative optimization process. This cycle is repeated multiple times, making it time-consuming and resource-intensive. ML holds promise to reduce these costs, speed up design cycles, and create better-performing designs [3, 31, 44, 22].

Yet, progress in ML methods for design is hindered by practical challenges. The first challenge is *limited data availability*. Since gathering label measurements is resource intensive [46, 28, 50, 19], designers are more often than not constrained to very small-scale datasets. In addition, there are often *non-smooth functional dependencies between the features and outcome*, complicating approximation, even for deep neural networks [30, 45]. Traditional methods use two part frameworks, requiring discriminators to guide the property enhancement for examples produced by a generative model. Such discriminators should be able to reliably predict the property of interest given some training data or its latent representations. Because of the dependency on training a discriminator for guidance,

---

[*] Prescient/MLDD, Genentech Research and Early Development
[†] Department of Computer Science, Center for Data Science, New York University

38th Conference on Neural Information Processing Systems (NeurIPS 2024).

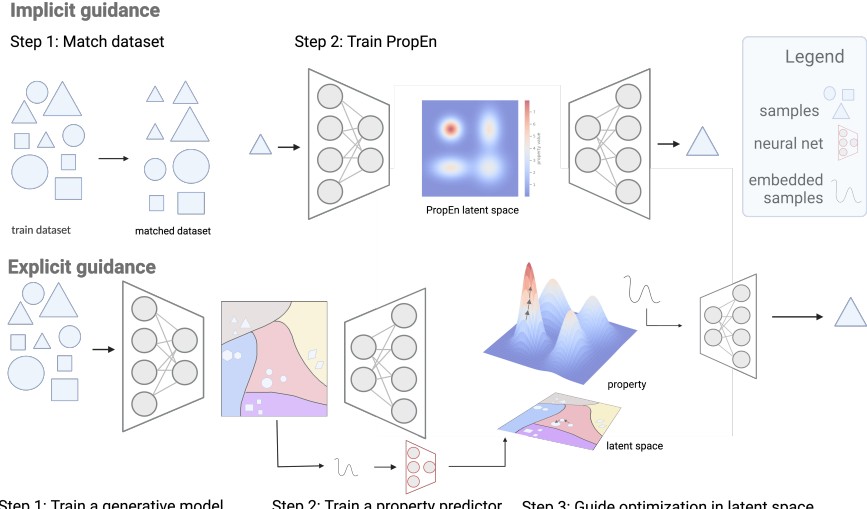

Figure 1: Conceptual summary of implicit and explicit guidance. The task is increasing the size of the objects. Top - in implicit guidance, first we match the training dataset, by pairing each sample with the closest one w.r.t. its shape, which has a better property (size). Then, we train a encoder-decoder framework which due to the construction of the dataset learns a lower dimensional manifold where the embeddings are ordered by the property value. Bottom - in explicit guidance, we train two separate models: a generator and a discriminator that guides the optimization in latent space.

we denote these methods as *explicit guidance*. While Genetic Algorithms were once prevalent [14], contemporary models like auto-encoders [47], GANs [49], and diffusion models now dominate both research and practice [31] the role of generative models. Despite their flexibility, such models face challenges typical to deep learning: they are "data-hungry" and unreliable when encountering out-of-distribution examples [29, 21, 7, 37].

Motivated by these challenges, we propose a new approach inspired by the concept of "matching". Matching techniques in econometrics are used to address the challenge of selection bias and confounding when estimating causal effects in observational studies [2, 38, 39, 43]. These techniques aim to create comparable groups of units by matching treated and control observations based on observable characteristics. The basic idea behind matching is to identify untreated units that are similar to treated units in terms of observed covariates, effectively creating a counterfactual comparison group for each treated unit. Matching techniques in ML as in econometrics have only been used to provide more robust, reliable causal-effect estimation [24, 48, 9].

This work argues that, in lack of large datasets, matching allows for implicit guidance, completely sidestepping the need for training a discriminator (differentiable surrogate model). We match each sample with a similar one that has a superior value for the property of interest. By doing so, we obtain a much larger training dataset, inherently embedding the direction of property enhancement. We name our method *PropEn* and we illustrate it in Figure 1. By leveraging this expanded dataset within a standard encoder-decoder framework, we circumvent the need for a separate discriminator model. We show that PropEn is domain agnostic, can be applied to any data modality continuous or discrete. Additionally, PropEn alleviates some common problems with explicit guidance such as falling off the data manifold or requiring complex engineering.

Overall, our contributions are as follows:

- We propose "matching" (inspired by causal effect estimation) to expand small training datasets (subsection 2.1);
- We provide a theoretical analysis on how training on a matched dataset implicitly learns an approximation of the gradient for a property of interest (subsection 2.2);

- We provide guarantees that the proposed designs are as likely as the training distribution, avoiding common pitfalls where unreliable discriminators lead to unrealistic, pathological designs (subsection 2.3);
- We demonstrate the effectiveness and advantages of implicit guidance through extensive experiments in both toy and real-world scientific problems, using both numerical and wet lab validation (section 3).

## 2 Property Enhancer (PropEn)

Given a set of initial examples, our objective is to propose a new design which is similar to the initial set, but, exceeds it in some property value of interest.

**Problem setup.** Concretely, we start with a dataset $\mathcal{D} = \{x_i, y_i\}_{i=1}^n$ consisting of $n$ observed examples $x_i \in \mathbb{R}^m$ drawn from a distribution $p$ together with their corresponding properties $y_i = g(x_i) \in \mathbb{R}$. Our objective is to determine ways to improve the property of a test example. Concretely, at test time, we are given a point $x_0$ and aim to identify some new point $x^{new} \sim p$ close to $x_0$ such that $g(x^{new}) > g(x_0)$. In effect, our problem combines constrained optimization (maximize $g(x^{new})$ while staying close to $x_0$) with sampling from a distribution (point $x^{new}$ should be likely according to $p$).

Hereafter, we will refer to the initial example we wish to optimize as *seed* design and the model's proposal as *candidate* design. Our method, *PropEn*, entails three steps: (i) matching a dataset, (ii) approximating the gradient by training a model with a matched reconstruction objective and (iii) sampling with implicit guidance.

### 2.1 Match the dataset

We view the group of samples with superior property values as the treated group and their lower-value counter part as the control group. This motivates us to construct a "matched dataset" for every $(x, y)$ within $\mathcal{D}$:

$$\mathcal{M} = \left\{ (x, x') \,\middle|\, \begin{array}{l} x, x' \in \mathcal{D} \\ \|x' - x\|^2 \leq \Delta_x, \ g(x') - g(x) \in (0, \Delta_y] \end{array} \right\}, \tag{1}$$

where $\Delta_x$ and $\Delta_y$ are predefined positive thresholds.

The matched dataset gives us a new and extended collection $\mathcal{M}$ whose size $N = O(n^2) \gg n$ can significantly exceed that of the training set, depending on the choice of matching thresholds.

### 2.2 Approximate the gradient

After matching the dataset, we train a deep encoder-decoder network $f_\theta$ over $\mathcal{M}$ by minimizing the *matched reconstruction* objective:

$$\ell(f_\theta; \mathcal{M}) = \frac{1}{|\mathcal{M}|} \sum_{(x,x') \in \mathcal{M}} \ell(f_\theta(x), x'), \qquad \text{(matched reconstruction objective)}$$

where $\ell$ is an appropriate loss for the data in question, such as an mean-squared error (MSE) or cross-entropy loss.

Before illustrating the properties of our method empirically, we perform a theoretical analysis. We show that minimizing the matched reconstruction objective yields a model that approximates the direction of the gradient of $g(\cdot)$, even if no property predictor has been explicitly trained:

**Theorem 1.** Let $f^*$ be the optimal solution of the matched reconstruction objective with a sufficiently small $\Delta_x$. For any point $x$ in the matched dataset for which $p$ is uniform within a ball of radius $\Delta_x$, we have $f^*(x) \to c\nabla g(x)$ for some positive constant $c$.

The detailed proof is provided in subsection A.1.

**Remark 1.** The proof of 1 is founded on the assumption that distribution is uniformly distributed within a ball of radius $\Delta_x$ around point $x$. This assumption is made to maintain the generality of the theorem without specific information about the sampling distribution, assuming uniformity avoids

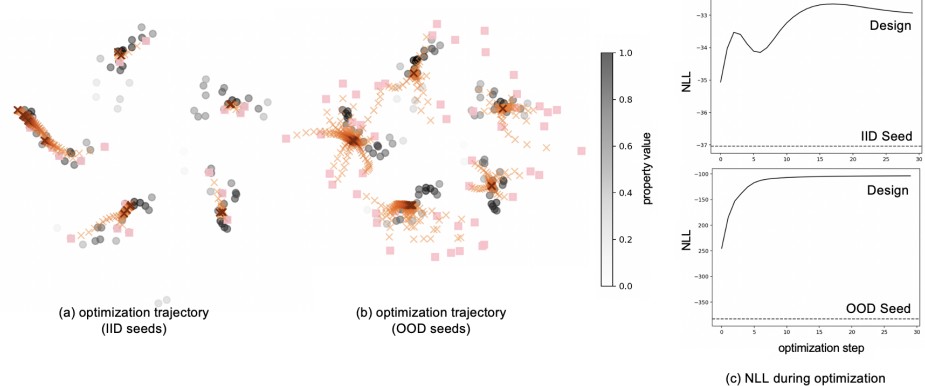

Figure 2: Illustration of PropEn on the pinwheel toy example with only 72 training examples. The training data are circles in grey, colored by the value of the property. With pink we mark the initial hold out test points and in orange '×' the PropEn trajectories. The color of the candidates intensifies with each iteration step. On the right-hand-side, we depict the sum of negative log likelihoods of the seeds and optimized designs across optimization steps.

introducing any biases that could arise from other distributional assumptions, such as symmetry, finite variance etc.

**Remark 2.** It can be shown that with with a matched reconstruction objective we learn a direction that is $a$-colinear with the gradient of $g$, avoiding the isotropy assumption. This leads to additional analysis on understanding the implications of the choices while matching, all included in Appendix A.

### 2.3 Optimize designs with implicit guidance

Training on a matched dataset allows for auto-regressive sampling. Starting with a design seed $x_0$, for $t = 1, 2, \ldots$, we can generate $x_t = f_\theta(x_{t-1})$ until convergence, $f_\theta(x_t) = x_t$, s.t. $g(x_t) > g(x_{t-1})$. At test time, we feed a seed design $x_0$ to PropEn, and read out an optimized design $x_1$ from its output. We then proceed to iteratively re-feed the current design to PropEn until $f_\theta(x_t) = x_t$, which is analogous to arriving at a stationary point with $\nabla g(x_t) = 0$ and we have exhausted the direction of property enhancement given the training data. Exploiting the implicit guidance from matching results in a trajectory of multiple optimized candidate designs.

We next show that optimized samples are almost as likely as our training set according to the data distribution $p$. This serves as a guarantee that the generated designs lie within distribution, as desired:

**Theorem 2.** Consider a model $f^*$ trained to minimize the matched reconstruction objective. The probability of $f^*(x)$ is at least

$$p(f^*(x)) \geq \mathbb{E}_{x' \sim \hat{\mu}_x}[p(x')] - \frac{\|H_p(f(x))\|_2 \, \sigma^2(\mathcal{M}_x)}{2},$$

where $\hat{\mu}_x$ is the empirical measure on the dataset, $H_p(x)$ is the Hessian of $p$ at $x$ and $\sigma^2(\mathcal{M}_x) = \mathbb{E}_{x' \sim \hat{\mu}_x}[\|x' - \mathbb{E}_{x'' \sim \hat{\mu}_x}[x'']\|_2^2]$ is the variance induced by the matching process.

The detailed proof is provided in subsection A.3.

We use a synthetic example to illustrate optimizing designs with PropEn. We choose a 2d pinwheel dataset. As a property to optimize, we choose the log-likelihood of the data as estimated by a KDE with Gaussian kernel with $\sigma = 0.01$. Figure 2 depicts in gray the training points, with the color intensity representing the value of the property—hence a higher/darker value is better in this example. After training PropEn, we take held out points (pink squares) and use them as seed designs. With orange x-markers, we illustrate PropEn candidates, with the color intensity increasing at each step $t$. We notice that PropEn moves towards the regions of the training data with highest property value, consistently improving at each step (right-most panel). Additionally, we also use out-of-distribution seeds, and we demonstrate in the middle panel that PropEn chooses to optimize them by proposing designs from the closest regions in the training data.

Table 1: Overview of the datasets in experiments.

| Dataset | Domain | Size $n$ | Type | Metric | Property | Preview |
|---------|--------|----------|------|--------|----------|---------|
| **Toy** | $\mathbb{R}^{10}, \mathbb{R}^{50}, \mathbb{R}^{100}$ | $50, 100$ | cont. | L2 | log-likelihood |  |
| **Airfoil** | $\mathbb{R}^{400}$ | $200, 500$ | cont. | L2 | lift-to-drag ratio |  |
| **Antibodies** | $20^{297}$ | $200 - 400$ | discrete | Levenshtein | binding affinity |  |

## 3 Experimental Results

We empirically evaluate PropEn on synthetic and real-world experiments to answer the following main questions: (i) Can PropEn be applied across various domains and datasets? (ii) Does PropEn provide reliable guidance, especially in situations with limited data and when dealing with out-of-distribution examples? (iii) How effective is PropEn in recommending optimal designs? Can it suggest candidates with property values exceeding those in the training set? (iv) How does PropEn's performance vary with different data characteristics (e.g., dimensionality, sample size, heterogeneity) and hyperparameters (such as $\Delta_x$, $\Delta_y$, and regularization terms)? Our code is available at https://github.com/prescient-design/propen.

**Datasets.** We consider three different data types: synthetic 2d toy datasets and their higher dimension transformations, NACA airfoil samples, and therapeutic antibody proteins. An overview of the data is given in Table 1. We present our results in two settings, *in silico* where we rely on experimental validation using computer simulations and solvers, and *in vitro* experiments where candidate designs were tested in a wet lab. Each of the experiments is evaluated under the baselines and metrics suitable for the domain.

**PropEn variants.** We investigate the utilization of matching and reconstruction within the PropEn framework. Two key considerations emerge: first, whether to reconstruct solely the input features (x2x) or both the input features and the property (xy2xy); second, the proximity to the initial sample, regulated by incorporating a straightforward reconstruction regularizer into the training loss $\ell(f_\theta(x), x)$. This regularized variant will be referred to as *mixup/mix*.

### 3.1 *In silico* experiments

#### 3.1.1 Toy data

We choose two well-known multi-modal densities: pinwheel and 8-Gaussians. These are 2d datasets, but, in order to make the task more challenging, we expand the dimensionality to $d \in \{10, 50, 100\}$ by randomly isometrically embedding the data within a higher dimensional space. Our findings are summarized in Figure 3 and we include the tabular results in item B.7. We empirically validate the four variants of PropEn and we compare against explicit guidance method: for consistency, we chose an auto-encoder of the same architecture as PropEn augmented with a discriminator for guidance in the latent space. We denote this baseline *Explicit*. We compare the methods by ratio of improvement , the proportion of holdout samples for which PropEn or baselines demonstrate enhanced property values. To assess the quality of the generated samples we report *uniqueness* and *novelty* in tables. We use a likelihood model derived from a Kernel Density Estimation (KDE) fit on the training data. The negative log-likelihood scores under this model serve as an indicator of in-distribution performance. Higher values for all metrics indicate better performance.

**Results.** Several insights can be gleaned from these experiments. When analyzing the results based on the number of samples, a clear trend emerges: as the number of training samples increases, PropEn consistently outperforms explicit guidance across all metrics, except for average improvement, where all methods exhibit similar behavior. The choice of the preferred metric may vary depending on the specific application; however, it is noteworthy that while explicit AE guidance improves approximately 50% of the designs for all datasets, PropEn demonstrates the potential to enhance up to 85% of the designs. Importantly, this improvement trend remains consistent regardless of the

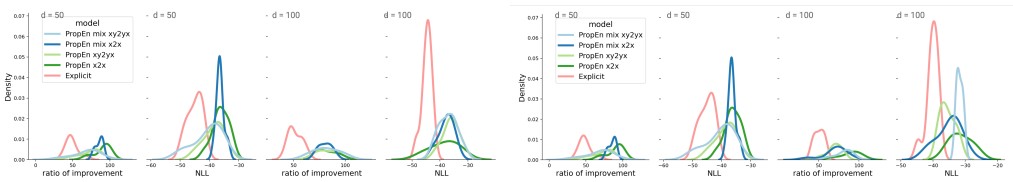

Figure 3: PropEn in toy examples in $d \in \{50, 100\}$, left side: 8-Gaussians, right side: pinwheel. Distribution of evaluation metrics from 10 repetitions of each experiment.

dimensionality of the data. Furthermore, an intriguing observation pertains to the performance of different variations of PropEn. It is noted that as the sample size increases, PropEn xy2xy does not exhibit an advantage over PropEn x2x. Moreover, the impact of iterative sampling with PropEn is notable. With each step, the property improves until it reaches a plateau after multiple iterations, albeit with a simultaneous drop in the uniqueness of the solution to around 80%. Nevertheless, iterative optimization can be continued until convergence, with all designs saved along the trajectory for later filtering according to user needs.

### 3.1.2   Engineering: airfoil optimization

In an engineering context, shape optimization entails altering the shape of an object to enhance its efficiency. NACA airfoils (National Advisory Committee for Aeronautics) [4], rooted in aerodynamics and parameterized by numerical values, serve as a well-documented benchmark due to their versatility. These airfoils span diverse aerodynamic characteristics, from high lift to low drag, making them ideal for exploring different optimization objectives. Given their integral role in aircraft wings, optimizing airfoil shapes can significantly impact aerodynamic performance by improving lift, drag, and other properties essential for aerospace engineering.

**Data & experimental design.** We generate NACA 4-digit series airfoil coordinates by choosing these parameters: $M$ (maximum camber percentage), $P$ (location of maximum camber percentage), and $T$ (maximum thickness percentage). Each airfoil is represented by 200 coordinates, resulting in a 400 vector representation when flattened. Our objective is to optimize the lift-to-drag ratio ($C_l/C_d$ ratio) for each shape. We calculate lift and drag using NeuralFoil [41], a precise deep-learning emulator of XFoil [11].

Note that the lift-to-drag ratio is pivotal in aircraft design, reflecting the wing's lift generation efficiency relative to drag production. A high value signifies superior lift production with minimal drag, translating to enhanced fuel efficiency, extended flight ranges, and overall improved performance. This ratio is paramount in aerodynamic design and optimization, facilitating aircraft to travel farther and more efficiently through the air. Traditionally, engineers have relied on genetic algorithms guided by Gaussian Process models (kriging) [22], however, in recent years the community has moved towards ML-based methods which consist of a generative model that can be GAN-based [51, 49] or a variation of a VAE [26, 52, 31, 47]. Similarly, for the surrogate, guidance model, GPs and numerical solvers have been replaced by deep models [5, 41]. For our experiments we follow this standard setup: we choose a VAE-like baseline as it is the most similar architectural choice to PropEn, and for guidance we use a MLP. All networks (encoder, decoder and surrogate) are fully connected 3 layer MLPs with 50 units, ReLU activations per layer and a latent space of dimension 50. We randomly select 0.1% as holdout dataset for seeds, and use the rest for training.

**Results.** Our numerical findings are summarized in **??**. Similar to the toy dataset, the designs produced by PropEn variants demonstrate enhanced properties compared to those guided explicitly. Delving into further analysis with ablation studies, depicted in Figure 4(b) and (c), we observe that increasing the matching thresholds in PropEn correlates with higher rates of improvement. Remarkably, all designs within a PropEn trajectory are deemed plausible, as depicted in the accompanying figure. Moreover, a consistent enhancement in the lift-to-drag ratio $C_l/C_d$ is noted along the optimization trajectory until convergence. This consistent trend underscores the effectiveness of PropEn in progressively refining airfoil designs to bolster their aerodynamic performance.

Interestingly, we find that in larger training datasets the threshold for property improvement ($\Delta_y$) may not be necessary for optimization, as the PropEn x2x demonstrates satisfactory performance.

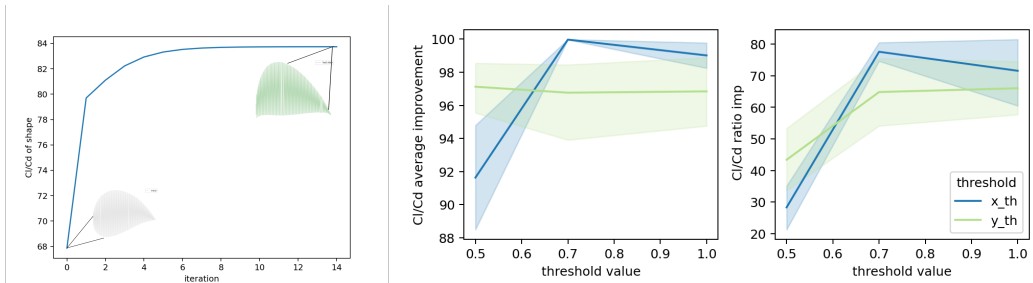

Figure 4: Ablation studies for PropEn on Airfoils. (a) PropEn improves Cl/Cd ratio along its trajectory and produces realistic/valid airfoil shapes. (b and c) the impact of choice of threshold $\Delta_x$ and $\Delta_y$ in the matching phase. When varying $\Delta_x$, $\Delta_y$ is set to 1, and vice versa.

We also notice that the mix variant of PropEn may require longer training. This observation is consistent with the notion that mix, by introducing a regularization term in the training loss, may improve at a slower rate compared to other PropEn variants. This slower improvement can be attributed to the regularization term's tendency to pull generated designs closer to the initial seed, thereby limiting the extent of exploration in the design space.

### 3.2 *In vitro* experiment: therapeutic protein optimization

The design of antibodies good functional and developability properties is essential for developing effective treatments for diseases ranging from cancer to autoimmune disorders. We here focus on the task of optimizing the binding affinity of a starting antibody (the seed) while staying close to it in terms of edit distance. The task is referred to as affinity maturation in the drug design literature and constitutes an essential and challenging step in any antibody design campaign.

Antibody binding affinity refers to the strength of the interaction between an antibody molecule and its target antigen. High binding affinity is crucial in antibody-based therapeutics as it determines the antibody's ability to recognize and bind to its target with high specificity and efficiency. We follow the standard practice of quantifying binding affinity by the negative log ratio of the association and dissociation constants (pKD), which represents the concentration of antigen required to dissociate half of the bound antibody molecules. Higher pKD indicates a tighter and more stable interaction, leading to improved therapeutic outcomes such as enhanced neutralization of pathogens or targeted delivery of drugs to specific cells.

**Data & experimental design.** The data collection process involved conducting low-throughput Surface Plasmon Resonance (SPR) experiments aimed at measuring with high accuracy the binding affinity of antibodies targeting three different target antigens: the human epidermal growth factor receptor 2 (HER2) and two additional targets that we denote as T1 and T2. For each of those targets, one or more seed designs were selected by domain experts. In the case of HER2, we used the cancer drug Herceptin as seed. We ensured the correctness of the SPR measurements by validating the fit of the SPR kinetic curves according to standard practices.

As the targets differ in the properties of their binding sites, we trained a PropEn model per each target (but for all seeds for that target jointly). For this application, we opted for the PropEn x2x mix variant. The reconstruction of the original sequence (mix) complies with antibody engineering wisdom that a candidate design should not deviate from a seed by more than small number of mutations. Similar to [34], we used a one-hot encoded representation of antibodies aligned according to the AHo numbering scheme [20] determined by ANARCI [12]. The encoder-decoder architecture is based on a ResNet [18] with 3 blocks each. We compare PropEn with four strong baselines: two state-of-the-art methods for guided and unguided antibody design namely walk-jump sampler [13] and lambo [17]; as well as two variants of a diffusion model trained on AHo antibody sequences differing on their use of guidance. The first one (labeled as **diffusion**) is based on a variational diffusion model [25] trained on a latent space obtained by projecting AHo 1-hot representation using an encoder-decoder type of architecture similar to *PropEn*'s architecture; encoder-decoder model is trained simultaneously with the diffusion model. The second one (labeled as **diffusion(guided)**) is a variant of the first one with

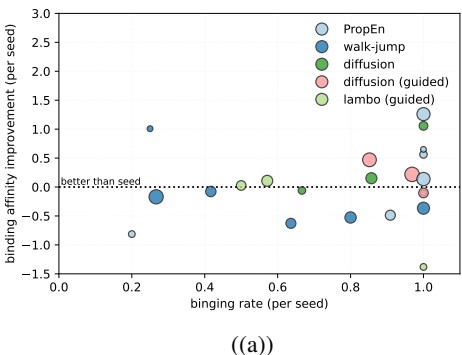
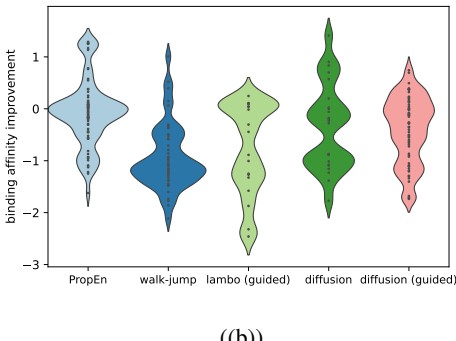

|  | ((a)) | ((b)) |
|---|---|---|

Figure 5: Therapeutic protein optimization results: (a) The left figure contrasts the binding rate with the 90-th percentile of the binding affinity improvement for each method and seed. Points on the top-right are on the Pareto front. (b) The right figure focuses on binders and reports the histograms of binding affinity improvement across all designs and seeds.

Table 2: Binding rate (and number of designs submitted). Higher is better.

|  | Herceptin | T1S1 | T1S2 | T1S3 | T2S1 | T2S2 | T2S3 | T2S4 | overall |
|---|---|---|---|---|---|---|---|---|---|
| **PropEn** | 90.9% (11) | 100.0% (4) | 100.0% (6) | 100.0% (24) | 20.0% (5) | 100.0% (23) | 100.0% (16) | 100.0% (4) | **94.6% (93)** |
| **walk-jump** [13] | - | 25.0% (4) | 80.0% (15) | 100.0% (18) | 26.7% (30) | 41.7% (12) | 100.0% (15) | 63.6% (11) | 62.9% (105) |
| **lambo (guided)** [17] | 50.0% (10) | 0.0% (4) | - | 100.0% (5) | 0.0% (9) | - | 100.0% (1) | 57.1% (14) | 44.2% (43) |
| **diffusion** | - | 100.0% (8) | 85.7% (14) | - | - | - | 88.2% (17) | 66.7% (6) | 86.7% (45) |
| **diffusion (guided)** | - | 85.2% (27) | 96.9% (32) | - | - | - | 93.3% (15) | 100.0% (10) | 92.9% (84) |

added guidance based on the *iterative latent variable refinement* idea described in the paper by Choi *et al.* [8], which ensures generating samples that are close to the initial seed.

We evaluate the set of designs in terms of their binding rate (fraction of designs tested that were binders), the percentage of designs than improve the seed, and their binding affinity improvement (pKD design - pKD of seed).

Table 3: Fraction of designs improving the seed and total designs tested. Higher is better.

|  | Herceptin | T1S1 | T1S2 | T1S3 | T2S1 | T2S2 | T2S3 | T2S4 | overall |
|---|---|---|---|---|---|---|---|---|---|
| **PropEn** | 0.0% (11) | 100.0% (4) | 33.3% (6) | 41.7% (24) | 0.0% (5) | 69.6% (23) | 0.0% (16) | 0.0% (4) | **34.4% (93)** |
| **walk-jump** [13] | - | 25.0% (4) | 6.7% (15) | 5.6% (18) | 3.3% (30) | 8.3% (12) | 0.0% (15) | 0.0% (11) | 4.8% (105) |
| **lambo (guided)** [17] | 10.0% (10) | 0.0% (4) | - | 0.0% (5) | 0.0% (9) | - | 0.0% (1) | 35.7% (14) | 14.0% (43) |
| **diffusion** | - | 62.5% (8) | 14.3% (14) | - | - | - | 0.0% (17) | 0.0% (6) | 15.6% (45) |
| **diffusion (guided)** | - | 51.9% (27) | 15.6% (32) | - | - | - | 0.0% (15) | 0.0% (10) | 22.6% (84) |

**Results.** As seen in Tables 2 and 3, PropEn excelled in generating functional antibodies with consistently high binding rates (94.6%) and 34.5% of the tested designs showed improve binding than the seed, outpacing other models in overall performance. To account for the trade-off between binding rate and affinity improvement (larger affinity improvement requires making risky mutations that might end-up killing binding), we visualize the Pareto front in Figure 5(a). In the plot, we mark the performance of each method for a specific seed design by placing a marker based on the achieved binding rate (x-axis) and maximum affinity improvement (y-axis). Compared to baseline methods, PropEn struck a beneficial trade-off, on average achieving a larger affinity improvement than methods with a high binding rate.

Figure 5(b) takes a closer look at the affinity improvement on the subset of designs that bound. As observed, all models produced some binders that were better than the seed, speaking for the strength of all considered models. Interestingly, none the top three models in terms of binding affinity improvement relied on explicit guidance, which aligns with our argument about the brittleness of explicit guidance in low-data regimes. Out of the those three models, PropEn generated two designs that improved the seed by at least one pKD unit (10 times better binder) followed by the walk-jump and the unguided diffusion model, that generated one such design each.

# 4  Related work

As design optimization has been ubiquitous across science domains, naturally our approach relates to a variety of methods and applications. In the molecular design domain, data are bound to discrete representation which can be challenging for ML methods. A natural way to circumvent that is by optimization in a latent continuous space. Bombarelli et al. [15] presented such an approach in their seminal work on optimizing molecules for chemical properties, and it has since spawn across different domains [6]. Recently, one of the common way for obtaining embeddings for explicit guidance relies on language models [27, 32]. One of the challenges of using a latent space is the issue for blindly guiding the optimization into ambiguous regions of the manifold where no training data was available [21, 29]. Follow up works attempt to address this problem [7, 16, 37] by incorporating uncertainty estimates into black-box optimization.

Another line of work perhaps more close to our approach is the notion of neural translation, where the goal is to go from one language to another by training on aligned datasets. [23, 10] have build on this idea to improve properties for small molecules translating one graph or sequence to another one with better properties. These works propose tailored approaches for domain specific applications. With PropEn we take a step further and propose a domain-agnostic framework that is empirically validated across different domains (uniquely including wetlab experiments). We also derive novel theoretical guarantees that illustrate the relation of the generated samples with the property gradient, as well as provide guarantees that our designs fall within distribution.

Previous works have also considered learning an optimizer for some function based on observed samples [42, 1, 36]. This is usually achieved by either (i) rendering the optimizer differentiable and training some of its hyperparameters; or (ii) by unfolding the iterative optimizer and treating each iteration as a trainable layer. Our approach is different, thanks to the matched reconstruction objective that lets us implicitly approximate the gradient of a function of interest.

# 5  Conclusion

**Strengths.** We introduced PropEn, a new method for implicit guidance in design optimization that approximates the gradient for a property of interest. We achieve this by leveraging matched datasets, which increase the size of the training data and inherently include the direction of property enhancement. Our findings highlight the versatility and effectiveness of PropEn in optimizing designs in engineering and drug discovery domains. We include wet lab in-vitro results for comparison with state-of-the-art baselines in therapeutic protein design. By utilizing thresholds for shape dissimilarity and property improvement, PropEn efficiently navigates the design space, generating diverse and high-performance configurations. We believe our method offers a simple yet effective recipe for design optimization that can be applied across various scientific domains.

**Limitations.** The matching step adds some computation overhead with complexity depending on the choice of distance metric. Since PropEn is targeting low data-regime applications, scalability is out of scope for the current work. However, we are considering on-the-fly distance evaluation or parallelisation across multiple nodes. The choice of distance metric for matching to a certain extent, can be considered a limitation because it requires understanding of the context for the application. However, this choice is also what allows for incorporating domain knowledge and constraints, which can be meaningful and necessary in the domain of interest (edit distance for antibodies, deviations only in the camber of the airfoil etc).

**Future work.** Immediate extensions of PropEn are applications to other molecular modalities, such as small molecules, material discovery, and optimization of adeno-associated virus vectors for gene-therapy. Additionally, we are keen to explore how different similarity metrics incorporate various inductive biases that can be leveraged for property optimization. Ongoing worthwhile efforts include developing a multi-property PropEn framework to address the optimization of properties simultaneously, offering a more comprehensive approach to the design process.

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

# Appendix / supplemental material

## A  Deferred proofs

### A.1  Proof of Theorem 1

*Proof.* The matched reconstruction objective for the MSE loss can be expressed as

$$\arg\min_{\theta} \sum_{x\sim\mathcal{X}} \sum_{\delta_x\sim U(0,\Delta_x)^m} \|f_\theta(x) - (x+\delta_x)\|^2\, \mathbb{1}(g(x) < g(x+\delta_x)). \tag{2}$$

Assuming a model that is sufficient overparametrized, we can w.l.o.g. suppose that there is some $\theta_x$ which minimizes every $x$ and thus swap the sum and argmin. We are left with the objective of minimizing the inner sum:

$$\arg\min_{\theta_x} \sum_{\delta_x\sim U(0,\Delta_x)^m} \|f_{\theta_x}(x) - (x+\delta_x)\|^2\, \mathbb{1}(g(x) < g(x+\delta_x)) \tag{3}$$

Noting that the the squared loss is minimized by the expected value and taking the data limit we get:

$$f_{\theta_x}(x) = \frac{1}{C} \sum_{\delta_x\sim\mathcal{U}(0,\Delta_x)^m} \delta_x\, \mathbb{1}(g(x) < g(x+\delta_x))$$

$$\xrightarrow[n\to\infty]{} \mathbb{E}_{\delta_x\sim\mathcal{U}(0,\Delta_x)^m}(\delta_x \mid g(x) < g(x+\delta_x)) \tag{4}$$

where constant $C$ is needed because the expected value should only be computed on the non-zero terms. To proceed, we consider a rotation matrix $R_x$ for which the following holds:

$$R_x\nabla g(x) = \begin{pmatrix} \|\nabla g(x)\|_2 \\ 0 \\ \vdots \\ 0 \end{pmatrix} \tag{5}$$

and introduce the reparametrization $z = R_x\delta_x$. The last vector is also uniformly distributed

$$z \sim R_x\mathcal{U}(0,\Delta_x)^m \sim \mathcal{U}(0,\Delta_x)^m$$

due to the uniform distribution on a ball being rotation invariant. Note that above we write $R_x\mathcal{U}(0,\Delta_x)^m$ to mean the push-forward of the distribution through the inverse rotation.

We can now rewrite Equation 4 as follows:

$$\mathbb{E}_{\delta_x\sim\mathcal{U}(0,\Delta_x)^m}(\delta_x)|g(x) < g(x+\delta_x) = R_x^{-1}\mathbb{E}_{\delta_x\sim\mathcal{U}(0,\Delta_x)^m}(R_x\delta_x) \mid g(x) < g(x+\delta_x) \tag{6}$$

$$= R_x^{-1}\mathbb{E}_{t\sim\mathcal{U}(0,\Delta_x)^m}t \mid g(x) < g(x+R_x^{-1}z) \tag{7}$$

$$= R_x^{-1}\mathbb{E}_{z\sim\mathcal{U}(0,\Delta_x)^m}z \mid 0 < \nabla g^\top(x)\,R_x^{-1}z + \epsilon(g,x,\Delta_x)) \tag{8}$$

Above, to go from line 7 to line 8 we Taylor expand $g$ around $x$:

$$g(x + R_x^{-1}z) = g(x) + \nabla g(x)^\top R_x^{-1}z + \epsilon(g,x,\Delta_x)$$

with $|\epsilon(g,x,\Delta_x)| = O(\Delta_x^2)L$ and $L$ being the Lipschitz constant of $\nabla g$.

The conditional in Equation 8 is equal to

$$-\epsilon(g,x,\Delta_x) < \nabla g(x)^\top R_x^{-1}z = z^\top R_x\nabla g^\top(x) = z^\top \begin{pmatrix} \|\nabla g(x)\|_2 \\ 0 \\ \vdots \\ 0 \end{pmatrix} \tag{9}$$

or equivalently

$$z_1 > \frac{-\epsilon(g,x,\Delta_x)}{\|\nabla g(x)\|_2}, \tag{10}$$

where we have denoted the first coordinate of $z$ as $z_1$.

Thus, setting $c = \frac{-\epsilon(g,x,\Delta_x)}{\|\nabla g(x)\|_2}$, Equation 8 can be re-written as

$$R_x^{-1}\mathbb{E}_{\delta_x \sim \mathcal{U}(0,\Delta_x)^m}(z|z_1 > c) = R_x^{-1}\begin{pmatrix} \mathbb{E}(z_1|z_1 > c) \\ \vdots \\ \mathbb{E}(z_m|z_1 > c) \end{pmatrix} \tag{11}$$

$$= \mathbb{E}(z_1|z_1 > c)\, R_x^{-1}\begin{pmatrix} 1 \\ 0 \\ \vdots \\ 0 \end{pmatrix} \tag{12}$$

$$= \frac{\mathbb{E}(z_1|z_1 > c)}{\|\nabla g(x)\|_2}R_x^{-1}R_x\nabla g(x) = \frac{\mathbb{E}(z_1|z_1 > c)}{\|\nabla g(x)\|_2}\nabla g(x). \tag{13}$$

We further note that $\mathbb{E}(z_1|z_1 > c) = (\Delta_x - \frac{-\epsilon(g,x,\Delta_x)}{\|\nabla g(x)\|_2})/2$ due to $z_1$ being distributed uniformly in $U(0, \Delta_x)$. We have thus shown that

$$f_{\theta_x}(x) \xrightarrow[n\to\infty]{} \frac{(\Delta_x + \frac{\epsilon(g,x,\Delta_x)}{\|\nabla g(x)\|_2})}{2\|\nabla g(x)\|_2}\nabla g(x), \tag{14}$$

and the constant in front of the gradient is positive for $\Delta_x > \frac{|\epsilon(g,x,\Delta_x)|}{\|\nabla g(x)\|_2}$. The latter condition is met whenever $\Delta_x = O(\frac{\|\nabla g(x)\|_2}{L})$. $\qquad\square$

## A.2 Understanding the relation between the learned direction and the property gradient

Our approach entails constructing a conditional '*matching distribution*':

$$\mu_x(x') \propto \begin{cases} p(x') & \text{if} \quad \|x' - x\|_2^2 \leq \Delta_x,\ g(x') - g(x) \in (\delta_y, \Delta_y] \\ 0 & \text{otherwise,} \end{cases} \tag{15}$$

and then training a model to optimize the following regularized matched reconstruction objective:

$$\ell(f, \hat{p}) = \mathbb{E}_{x \sim \hat{p}}[\mathbb{E}_{x' \sim \hat{\mu}_x}[\ell(x', f(x)) + \beta\,\ell(x, f(x))]].$$

Note that by $\hat{p}(x)$ we denote the empirical density supported on the training set $\hat{p}(x) = \frac{1}{n}\sum_{i=1}^n \delta(x_i - x)$ (and analogously for $\hat{\mu}_x$).

**Theorem 3.** Let $f^*$ be the optimal solution of the matched reconstruction objective matched reconstruction objective. For any point $x$, the global minimizer is given by

$$f^*(x) = \frac{\mathbb{E}_{x' \sim \hat{\mu}_x}[x'] + \beta x}{1 + \beta}. \tag{16}$$

Further, for a $\lambda_1$-Lipschitz and $\lambda_2$-smooth function $g$, the vector $f^*(x) - x$ is $a$-colinear with the gradient of $g$:

$$a \leq \frac{\nabla g(x)^\top (f^*(x) - x)}{\|\nabla g(x)\|_2 \|f^*(x) - x\|_2}$$

whenever $\Delta_x < 2\frac{\delta_y - \alpha\lambda_1\|\mathbb{E}_{x' \sim \hat{\mu}_x}[x'] - (1-\beta)x\|_2}{\lambda_2}$.

*Proof.* The global minimizer of the matched reconstruction objective for the MSE loss is given by

$$f^*(x) = \frac{\mathbb{E}_{x' \sim \hat{\mu}_x}[x'] + \beta x}{1 + \beta} \tag{17}$$

which directly follows by taking the gradient of the mean-squared error loss and setting it to zero.

We next consider a $C^2$ function $g$ and Taylor expand it around $x$:

$$g(x') = g(x) + \nabla g(x)^\top (x' - x) + \epsilon(x, x'),$$

where the norm of the approximation error is at most $|\epsilon(x, x')| \leq \lambda_2 \|x - x'\|_2^2/2 \leq \lambda_2 \Delta_x/2$.

Taking the expectation w.r.t. $\hat{\mu}_x$ yields

$$\mathbb{E}_{x' \sim \hat{\mu}_x}[g(x')] - g(x) = \nabla g(x)^\top (\mathbb{E}_{x' \sim \hat{\mu}_x}[x'] - x) + \mathbb{E}_{x' \sim \hat{\mu}_x}[\epsilon(x, x')]$$

$$\Leftrightarrow \mathbb{E}_{x' \sim \hat{\mu}_x}[g(x') - \epsilon(x, x')] - g(x) = \nabla g(x)^\top (f^*(x) - x)(1 + \beta),$$

where in the last step we substituted the expectation by $f'(x)$. We notice that, as long as $\mathbb{E}_{x' \sim \hat{\mu}_x}[g(x') - \epsilon(x, x')] - g(x) > 0$, the learned direction is pointing towards a similar direction as the gradient:

$$\angle(\nabla g(x), f^*(x) - x) = \arccos\left(\frac{\nabla g(x)^\top (f^*(x) - x)}{\|\nabla g(x)\|_2 \|f^*(x) - x\|_2}\right) \leq 90° \tag{18}$$

We expand the term within the condition in the following:

$$\mathbb{E}_{x' \sim \hat{\mu}_x}[g(x') - \epsilon(x, x')] - g(x) \geq \inf_{x'} \mathbb{E}_{x' \sim \hat{\mu}_x}[g(x') - |\epsilon(x, x')|] - g(x)$$

$$\geq \delta_y - \frac{\lambda_2 \Delta_x}{2}$$

to determine that the following sufficient condition for $\angle(\nabla g(x)^\top, f^*(x) - x)$ to be below 90 degrees:

$$\Delta_x < \frac{2\delta_y}{\lambda_2}.$$

More generally, since

$$\frac{\nabla g(x)^\top (f^*(x) - x)}{\|\nabla g(x)^\top\| \|f^*(x) - x\|} \geq \frac{\delta_y - \frac{\lambda_2 \Delta_x}{2}}{(1 + \beta)\lambda_1 \|f^*(x) - x\|_2}$$

a sufficient condition for the normalized inner product to be above $\alpha$ is

$$a \leq \frac{\nabla g(x)^\top (f^*(x) - x)}{\|\nabla g(x)^\top\| \|f^*(x) - x\|} \Leftarrow \Delta_x < 2 \frac{\delta_y - \alpha(1 + \beta)\lambda_1 \|f^*(x) - x\|_2}{\lambda_2}$$

One may also set $\|f^*(x) - x\|_2 = \frac{\mathbb{E}_{x' \sim \hat{\mu}_x}[x'] - (1 - \beta)x}{1 + \beta}$ in the equation above to obtain the condition

$$\Delta_x < 2 \frac{\delta_y - \alpha\lambda_1 \|\mathbb{E}_{x' \sim \hat{\mu}_x}[x'] - (1 - \beta)x\|_2}{\lambda_2}.$$

as claimed. □

## A.3 Proof of Theorem 2

*Proof.* Set $x' = f^*(x)$. Taking a Taylor expansion of $p$ around $x'$, we deduce that for every $x'' \in D$ the following holds:

$$p(x_i) \leq p(x') + \nabla p(x')^\top (x'' - x) + \frac{\|H_p(x')\|_2 \|x'' - x\|_2^2}{2},$$

with $\|H_p(x')\|_2$ being the Hessian of $p$ at $x'$. Taking the expectation w.r.t. $\hat{\mu}_x$ yields

$$\mathbb{E}_{x'' \sim \hat{\mu}_x}[p(x'')] \leq p(x') + \nabla p(x')^\top (\mathbb{E}_{x'' \sim \hat{\mu}_x}[x''] - x') + \frac{\|H_p(x')\|_2 \mathbb{E}_{x'' \sim \hat{\mu}_x}[\|x'' - x'\|_2^2]}{2}$$

or equivalently

$$p(x') \geq \mathbb{E}_{x'' \sim \hat{\mu}_x}[p(x'')] - \nabla p(x')^\top (\mathbb{E}_{x'' \sim \hat{\mu}_x}[x''] - x') - \frac{\|H_p(x')\|_2 \mathbb{E}_{x'' \sim \hat{\mu}_x}[\|x'' - x'\|_2^2]}{2}$$

If we further assume that the model is trained to minimize the matched reconstruction objective with an MSE loss, we obtain the claimed result:

$$p(x') \geq \mathbb{E}_{x'' \sim \hat{\mu}_x}[p(x'')] - \frac{\|H_p(x')\|_2 \sigma^2(\mathcal{M}_x)}{2},$$

where $\sigma^2(\mathcal{M}_x) = \mathbb{E}_{x' \sim \hat{\mu}_x}[\|x' - \mathbb{E}_{x'' \sim \hat{\mu}_x}[x'']\|_2^2]$ is the variance induced by the matching process. □

Observe that, since the variance is upper bounded by $\sigma^2(\mathcal{M}_x) \leq 4\Delta_x$, one may provide higher likelihood samples by restricting the matching process to consider closer pairs. Further, as perhaps expected, the likelihood is higher for smoother densities.

### A.4 Understanding the matched reconstruction objective

The following analysis analyses the key characteristics of the proposed method. Though our analysis does not take into account the training process with stochastic gradient descent and the potentially useful inductive biases of the neural network $f_\theta$, it provides useful insights that qualitatively align with the practical model behavior.

**Property 1. The learned direction follows the property gradient.** We first show that by optimizing the matched reconstruction objective, the model will learn to adjust an input point by approximately following the gradient $\nabla g(x)$ of the property function.

For ease of notation, we denote by $\mathcal{M}_x$ the subset of our paired dataset with all pairs that start from point $x$.

**Theorem 4.** Let $f_\beta^*$ be the optimal solution of the matched reconstruction objective. For a $\lambda_1$-Lipschitz and $\lambda_2$-smooth function $g$, the vector $v_x = f_\beta^*(x) - x$ is $a$-colinear with the gradient of $g$:

$$a \le \frac{\nabla g(x)^\top v_x}{\|\nabla g(x)\|_2 \|v_x\|_2}$$

whenever $\Delta_x < \frac{2}{\lambda_2} \left( \delta_y - \alpha \lambda_1 (1 + \beta) \|f_\beta^*(x) - x\|_2 \right)$.

The theorem reveals the role of the introduced thresholds in controlling how close we follow the property gradient: the larger the required property change $\delta_y$ and the smaller the allowed input deviation $\Delta_x$ the closer $v_x$ tracks the gradient of $g$. We thus observe a trade-off between approximation and dataset size: selecting less conservative thresholds will lead to a larger $|\mathcal{M}|$ at the expense of a rougher approximation.

**Property 2. Regularization controls the step size.** Hyperparameter $\beta$ enables us to control the magnitude of the allowed input change (the step size in optimization parlance), with larger $\beta$ encouraging smaller changes:

**Corollary 1.** In the setting of Theorem 4, we have: $\|f_\beta^*(x) - x\|_2 = \frac{\|f_0^*(x) - x\|_2}{1+\beta}$.

Therefore, setting $\beta$ to a larger value can be handy for conservative design problems that prioritize distance constraints or when the property function is known to be complex.

## B  Additional results for experiments

### B.1  Details on Matched datasets

Table 4: Overview of matched dataset and choice of $\Delta_x$ and $\Delta_y$.

| Dataset | # Pairs | # Unique Samples in Train | Y Range (Control) | Y Range (Treatment) |
|---|---|---|---|---|
| 8 Gaussians | 1 746 | 96 | [0.14, 0.86] | [0, 1] |
| Airfoils | 8 125 | 200 | [56.3, 82.8] | [65.2, 90.6] |
| Antibodies (T1) | 1 362 | 268 | [4, 8.9] | [6.5, 9.3] |

### B.2  Toy experiment - 8 Gaussians with anti-clockwise increasing property value

We include a toy example where the property is disentangled from the likelihood of the data. The results in Figure B.2 and Figure B.2 are consistent with the discussion in the main text.

### B.3  Setting $\Delta_x$ and $\Delta_y$

The choice of parameters $\Delta_x$ and $\Delta_y$ should be informed by the specific application. For example, in antibody design, domain experts recommend not using thresholds above a Levenshtein distance of 8, as such differences are considered biologically irrelevant. Similarly, for $\Delta_y$, knowing that noise in binding measurements can be up to 0.3, we chose 0.5 to ensure proper matching. When faced

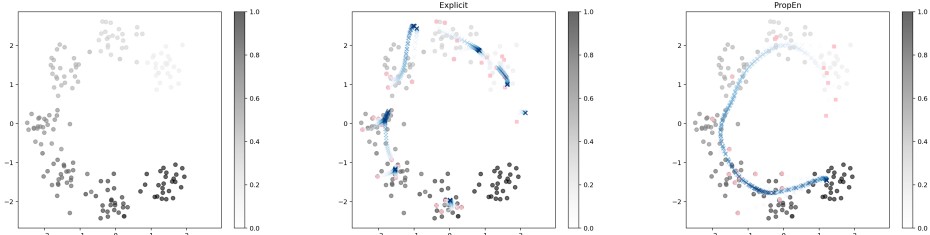

Figure 6: **New toy example** $d = 50$. Explicit guidance suffers in higher dimensions (shown here for $d = 50$) as there are increasingly more directions around the data in which the surrogate model is erroneous. Implicit guidance is robust to the higher data dimensionality because generated samples lie within distribution (as proven).

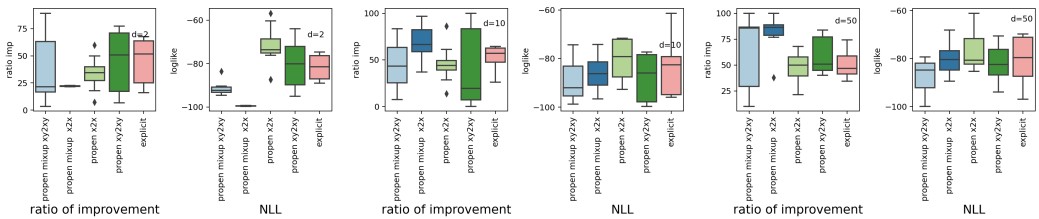

Figure 7: New toy example in $d \in \{2, 10, 50\}$, for 8-Gaussians with property increasing in anti-clockwise direction. Distribution of evaluation metrics from 10 repetitions of each experiment. ratio of improvement, higher is better, NLL lower is better.

with a new dataset of $n$ unique data points, a good initial approach is to use the standard deviation of pairwise distances for $x$, and the mean or median for property $y$:

$$\Delta_x \sigma_d = \sqrt{\frac{2}{n(n-1)} \sum_{1 \leq i < j \leq n} \left( d(x_i, x_j) - \frac{2}{n(n-1)} \sum_{1 \leq k < l \leq n} d(x_k, x_l) \right)^2}$$

$$\Delta_y \leq \frac{2}{n(n-1)} \sum_{1 \leq i < j \leq n} d(x_i, x_j)$$

**How $g(\cdot)$ influences the calibration/selection of $\Delta_y$**
When the $g(.)$ is non-smooth, or we have a very few datapoints, this certainly influences the choice of $\Delta_y$, in this case we opt for as small steps as possible since this will create many pairs which should give sense of the direction of the gradient around the sparse regions.

In the Figure 8, we provide an additional ablation study on how threshold choices affect the number of training pairs, showing that sufficiently large thresholds include all unique training points. For performance impact, please refer to the ablation study in Figure 4.

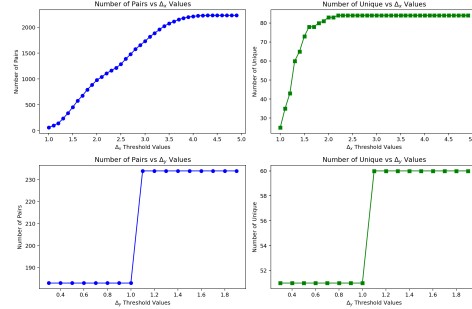

Figure 8: Effect of thresholds $\Delta_x$ and $\Delta_y$ on the number of pairs. Though the number of pairs increases with larger thresholds, the benefit saturates (especially when increasing $\Delta_y$).

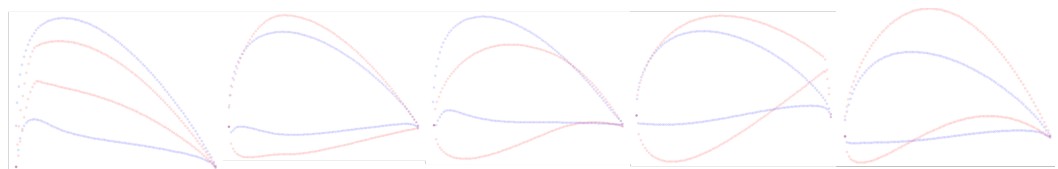

Figure 9: Airfoils optimized with PropEn from multiple seeds. Red: seed designs, blue: PropEn candidates. To reduce the lift-drag ratio, PropEn tends to flatten the bottom of the airfoil to reduce the drag, while extend the front upwards to increase lift.

Table 5: Diffusion models do improve almost all hold out designs (RI - ratio of improvement), however, only by a very small value (AI-average improvement.)

| Airfoil experiment $N = 200$ | AI | RI |
|---|---|---|
| **Explicit guidance** | $15.04 \pm 1.21$ | $6.53 \pm 6.35$ |
| **Diffusion** $T = 5$ | $4.297 \pm 0.7$ | $97.6 \pm 0.3$ |
| **Diffusion** $T = 15$ | $4.54 \pm 0.3$ | **99.2** $\pm 0.2$ |
| **Diffusion** $T = 150$ | $4.42 \pm 0.3$ | $97.38 \pm 0.4$ |
| **PropEn mix x2x** | $29.49 \pm 8.02$ | $7.23 \pm 6.21$ |
| **PropEn x2x** | $41.30 \pm 14.83$ | $38.83 \pm 8.80$ |
| **PropEn mix xy2xy** | $5.91 \pm 8.72$ | $6.06 \pm 7.75$ |
| **PropEn xy2xy** | **55.41** $\pm 14.07$ | $29.81 \pm 31.30$ |

## B.4 Airfoils

### B.4.1 Additional baseline for airfoils

## B.5 Example of PropEn designs for antibodies

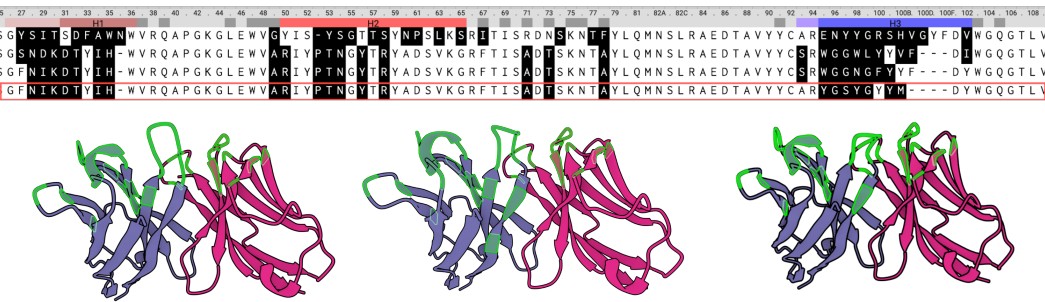

Figure 10: HER2 binders from PropEn validated in wet lab experiments. Up: sequence alignment of the heavy chains wrt the seed which is the top sequence. The positions marked in black correspond to mutational differences from the seed. Bottom: folded structures of the corresponding designs with the mutations to seed marked in green.

## B.6 Details on experimental setups

**Toy datasets**

- ablation studies: $N \in \{100, 200\}, d \in \{2, 10, 50, 100\}$
- matching thresholds: $\Delta_x = \Delta_y = 1$,
- number of epochs: 500, batch size: 64

**Criterion**: MSELoss()

**Encoder**:

```
Sequential(
    (0): Linear(in_features=2, out_features=30, bias=True)
    (1): ReLU()
    (2): Linear(in_features=30, out_features=30, bias=True)
    (3): ReLU()
    (4): Linear(in_features=30, out_features=30, bias=True)
    (5): ReLU()
    (6): Linear(in_features=30, out_features=15, bias=True)
)
```

**Decoder**: Analogous to the encoder, starting with the compressed representation (15 features) and progressively reconstructing the original input size (2 features).

**Toy Example Discriminator:**

```
Sequential(
    (0): Linear(in_features=2, out_features=30, bias=True)
    (1): ReLU()
    (2): Linear(in_features=30, out_features=30, bias=True)
    (3): ReLU()
    (4): Linear(in_features=30, out_features=30, bias=True)
    (5): ReLU()
    (6): Linear(in_features=30, out_features=1, bias=True)
)
```

**Airfoil**

- ablation studies: $N \in \{200, 500, 1000\}$,
- matching thresholds: $\Delta_x = \Delta_y \in \{0.3, 0.5, 0.7, 1\}$
- number of epochs: 1000, batch size: 100

**Criterion**: `MSELoss()`

**Encoder**:

```
Sequential(
    (0): Linear(in_features=2, out_features=100, bias=True)
    (1): ReLU()
    (2): Linear(in_features=100, out_features=100, bias=True)
    (3): ReLU()
    (4): Linear(in_features=100, out_features=100, bias=True)
    (5): ReLU()
    (6): Linear(in_features=100, out_features=50, bias=True)
)
```

**Decoder**: Analogous to the encoder, starting with the compressed representation (50 features) and progressively reconstructing the original input size (400 features).

**AirFoil Discriminator:**

```
Sequential(
    (0): Linear(in_features=100, out_features=100, bias=True)
    (1): ReLU()
    (2): Linear(in_features=100, out_features=100, bias=True)
    (3): ReLU()
    (4): Linear(in_features=100, out_features=100, bias=True)
    (5): ReLU()
    (6): Linear(in_features=100, out_features=1, bias=True)
)
```

**Hyper-parameter choice.**  For optimizing the parameters of the baselines in the toy and engineering experiments, we conducted a grid search over the learning rate ([1e-2, 1e-5]), weight decay ([1e-2, 1e-5]), number of epochs ([300, 1000, 5000]), batch size (32, 64, 128), and number of neurons per layer ([30, 50, 100]).

**Therapeutic Proteins**

- batch size: 32, training epochs: 300
- matching thresholds: $\Delta_x = \in \{10, 15\}$ mutational, edit distance, $\Delta_y = 0.5$
- baselines: For WJS, we used a 1D Conv architecture (please see A.1 in [38]), and for Lambo, a bert-small transformer (please see C.1 in [39]).

```
Encoder(
   (mlp): Linear(in_features=6556, out_features=128, bias=True)
   (blocks): ModuleList(
     3 x ResNetBlock(
        (ln1): LayerNorm((128,), eps=1e-05, elementwise_affine=True)
        (mlp1): Linear(in_features=128, out_features=256, bias=True)
        (ln2): LayerNorm((256,), eps=1e-05, elementwise_affine=True)
        (mlp2): Linear(in_features=256, out_features=128, bias=True)
        (act): GELU(approximate='none')
     )
```

**Decoder**: Analogous to the encoder, starting with the compressed representation (256 features) and progressively reconstructing the original input size (6556).

**Comparison to Bayesian Optimization.**  While both BO and PropEn aim at optimization in small sample sizes, the two frameworks solve different problems. The goal of PropEn is to generate designs, whereas in BO/AL the goal is to choose the most promising designs that should be labeled in order to imporve a predictors performance or find the best candidate, i.e. the focus is selection. In the context of optimizing designs, one would have a suite of (1) generative models, (2) property predictors and (3) BO/AL module that will do the final selection across pool of candidates. PropEn falls in (1), the category of generative models section that will contribute to the library of potential candidates. As a side note, Lambo, a method we compared to, uses a BO inspired acquisition function to guide the search for better designs, and we do compare to it (favorably), but, we must highlight the difference, **PropEn and Lambo are generative models not BO/AL methods. We hope this clarifies the differences between the two frameworks and highlights their complementarity.

## B.7  Ablation study for toy experiments

Additional metrics included:

1. Uniqueness: number of unique designs divided by number of generated designs.
2. Novelty: number of designs proposed by the method that don't appear in the training data, divided by the number of generated designs.

Table 6: PropEn vs Explicit guidance results on toy datasets, end of optimization trajectory. $N$ is the number of samples, $d$ is the number of dimensions for projection (2D $\rightarrow d$D datasets). Mean (std) over 10 repetitions of the experiment. For all metrics, higher is better.

| | N=100 | | | N=200 | | |
|---|---|---|---|---|---|---|
| | d=10 | d=50 | d=100 | d=10 | d=50 | d=100 |
| **8-Gaussians** | | | | | | |
| **ratio imp** | | | | | | |
| PropEn mix x2x | $85.71 \pm 5.01$ | $77.89 \pm 16.79$ | $81.58 \pm 12.58$ | $92.82 \pm 5.64$ | $93.16 \pm 6.66$ | $91.88 \pm 8.49$ |
| PropEn mix xy2xy | $68.42 \pm 18.23$ | $65.41 \pm 9.04$ | $80.70 \pm 6.37$ | $79.49 \pm 10.44$ | $75.78 \pm 13.27$ | $79.49 \pm 13.57$ |
| PropEn x2x | $90.06 \pm 11.30$ | $91.81 \pm 10.23$ | $85.09 \pm 11.25$ | $93.33 \pm 4.87$ | $94.36 \pm 5.64$ | $93.77 \pm 5.51$ |
| PropEn xy2xy | $73.03 \pm 13.91$ | $79.70 \pm 11.94$ | $82.11 \pm 9.56$ | $89.01 \pm 5.87$ | $78.39 \pm 6.92$ | $79.77 \pm 10.76$ |
| Explicit | $48.80 \pm 8.52$ | $49.76 \pm 5.45$ | $51.67 \pm 6.58$ | $48.02 \pm 4.88$ | $48.02 \pm 4.15$ | $50.58 \pm 5.01$ |
| **loglike** | | | | | | |
| PropEn mix x2x | $-36.34 \pm 1.28$ | $-37.06 \pm 2.53$ | $-37.08 \pm 3.22$ | $-77.42 \pm 2.53$ | $-77.61 \pm 4.25$ | $-77.50 \pm 2.68$ |
| PropEn mix xy2xy | $-40.51 \pm 4.90$ | $-43.51 \pm 5.76$ | $-37.71 \pm 2.67$ | $-80.17 \pm 3.19$ | $-83.16 \pm 2.20$ | $-82.52 \pm 3.73$ |
| PropEn x2x | $-36.13 \pm 2.68$ | $-35.12 \pm 2.76$ | $-36.04 \pm 2.91$ | $-76.95 \pm 2.70$ | $-76.80 \pm 4.83$ | $-77.55 \pm 3.27$ |
| PropEn xy2xy | $-38.51 \pm 3.44$ | $-38.70 \pm 2.18$ | $-38.47 \pm 5.12$ | $-79.00 \pm 3.67$ | $-81.79 \pm 4.87$ | $-82.92 \pm 4.05$ |
| Explicit | $-45.00 \pm 2.50$ | $-44.11 \pm 2.46$ | $-44.64 \pm 1.71$ | $-99.34 \pm 3.02$ | $-99.62 \pm 3.84$ | $-98.88 \pm 3.63$ |
| **uniqness** | | | | | | |
| PropEn mix x2x | $93.98 \pm 9.33$ | $97.37 \pm 3.72$ | $89.47 \pm 12.26$ | $82.05 \pm 7.15$ | $77.78 \pm 6.01$ | $78.20 \pm 12.00$ |
| PropEn mix xy2xy | $96.84 \pm 5.08$ | $98.50 \pm 3.98$ | $93.86 \pm 3.96$ | $81.09 \pm 10.16$ | $84.62 \pm 8.79$ | $86.08 \pm 6.08$ |
| PropEn x2x | $79.53 \pm 10.67$ | $74.27 \pm 14.76$ | $77.19 \pm 14.38$ | $57.69 \pm 8.48$ | $57.95 \pm 12.69$ | $56.04 \pm 8.69$ |
| PropEn xy2xy | $78.95 \pm 9.75$ | $72.93 \pm 10.71$ | $81.05 \pm 16.48$ | $65.57 \pm 17.19$ | $55.68 \pm 8.34$ | $54.70 \pm 11.75$ |
| Explicit | $100.00 \pm 0.00$ | $100.00 \pm 0.00$ | $100.00 \pm 0.00$ | $100.00 \pm 0.00$ | $100.00 \pm 0.00$ | $100.00 \pm 0.00$ |
| **novelty** | | | | | | |
| PropEn mix x2x | $100.00 \pm 0.00$ | $100.00 \pm 0.00$ | $100.00 \pm 0.00$ | $100.00 \pm 0.00$ | $100.00 \pm 0.00$ | $100.00 \pm 0.00$ |
| PropEn mix xy2xy | $100.00 \pm 0.00$ | $100.00 \pm 0.00$ | $100.00 \pm 0.00$ | $100.00 \pm 0.00$ | $100.00 \pm 0.00$ | $100.00 \pm 0.00$ |
| PropEn x2x | $100.00 \pm 0.00$ | $100.00 \pm 0.00$ | $100.00 \pm 0.00$ | $100.00 \pm 0.00$ | $100.00 \pm 0.00$ | $100.00 \pm 0.00$ |
| PropEn xy2xy | $100.00 \pm 0.00$ | $100.00 \pm 0.00$ | $100.00 \pm 0.00$ | $100.00 \pm 0.00$ | $100.00 \pm 0.00$ | $100.00 \pm 0.00$ |
| Explicit | $100.00 \pm 0.00$ | $100.00 \pm 0.00$ | $100.00 \pm 0.00$ | $100.00 \pm 0.00$ | $100.00 \pm 0.00$ | $100.00 \pm 0.00$ |
| **pinwheel** | | | | | | |
| **ratio imp** | | | | | | |
| PropEn mix x2x | $86.73 \pm 10.56$ | $84.02 \pm 13.94$ | $80.94 \pm 10.80$ | $87.55 \pm 10.41$ | $91.35 \pm 7.75$ | $86.41 \pm 5.68$ |
| PropEn mix xy2xy | $71.56 \pm 13.09$ | $66.32 \pm 9.56$ | $70.39 \pm 15.13$ | $56.41 \pm 15.88$ | $51.57 \pm 20.53$ | $61.54 \pm 12.33$ |
| PropEn x2x | $89.29 \pm 10.09$ | $85.97 \pm 9.49$ | $89.47 \pm 12.89$ | $91.03 \pm 9.54$ | $92.67 \pm 6.69$ | $87.75 \pm 8.48$ |
| PropEn xy2xy | $71.67 \pm 12.14$ | $65.13 \pm 17.10$ | $69.67 \pm 9.04$ | $52.38 \pm 19.46$ | $47.86 \pm 16.98$ | $63.40 \pm 17.54$ |
| Explicit | $52.92 \pm 5.66$ | $49.55 \pm 5.14$ | $50.45 \pm 6.12$ | $51.75 \pm 4.70$ | $50.12 \pm 4.64$ | $50.35 \pm 3.85$ |
| **loglike** | | | | | | |
| PropEn mix x2x | $-31.48 \pm 2.70$ | $-31.44 \pm 2.62$ | $-31.73 \pm 0.96$ | $-68.34 \pm 3.31$ | $-67.43 \pm 2.70$ | $-69.00 \pm 2.71$ |
| PropEn mix xy2xy | $-34.25 \pm 2.13$ | $-35.33 \pm 1.26$ | $-35.06 \pm 3.84$ | $-83.83 \pm 8.83$ | $-86.89 \pm 11.47$ | $-80.11 \pm 4.17$ |
| PropEn x2x | $-31.47 \pm 1.36$ | $-31.12 \pm 2.20$ | $-31.37 \pm 3.44$ | $-68.18 \pm 2.88$ | $-66.66 \pm 1.96$ | $-68.12 \pm 2.92$ |
| PropEn xy2xy | $-34.96 \pm 3.91$ | $-37.10 \pm 4.35$ | $-35.50 \pm 2.29$ | $-87.73 \pm 8.75$ | $-88.06 \pm 13.58$ | $-79.83 \pm 7.58$ |
| Explicit | $-40.40 \pm 1.93$ | $-39.58 \pm 2.25$ | $-40.14 \pm 1.80$ | $-90.32 \pm 5.00$ | $-88.90 \pm 3.16$ | $-87.91 \pm 4.63$ |
| **uniqness** | | | | | | |
| PropEn mix x2x | $92.76 \pm 7.93$ | $94.67 \pm 4.65$ | $97.89 \pm 4.71$ | $80.59 \pm 12.19$ | $78.85 \pm 7.48$ | $82.56 \pm 10.31$ |
| PropEn mix xy2xy | $94.08 \pm 10.69$ | $96.84 \pm 2.88$ | $97.37 \pm 5.63$ | $98.90 \pm 2.02$ | $96.01 \pm 3.87$ | $90.51 \pm 10.54$ |
| PropEn x2x | $66.96 \pm 9.44$ | $68.42 \pm 15.57$ | $71.58 \pm 15.16$ | $49.23 \pm 15.84$ | $47.99 \pm 12.97$ | $50.71 \pm 8.39$ |
| PropEn xy2xy | $84.70 \pm 7.20$ | $72.37 \pm 17.29$ | $78.86 \pm 8.48$ | $69.96 \pm 9.21$ | $60.68 \pm 17.14$ | $64.80 \pm 10.58$ |
| Explicit | $100.00 \pm 0.00$ | $100.00 \pm 0.00$ | $100.00 \pm 0.00$ | $100.00 \pm 0.00$ | $100.00 \pm 0.00$ | $100.00 \pm 0.00$ |
| **novelty** | | | | | | |
| PropEn mix x2x | $100.00 \pm 0.00$ | $100.00 \pm 0.00$ | $100.00 \pm 0.00$ | $100.00 \pm 0.00$ | $100.00 \pm 0.00$ | $100.00 \pm 0.00$ |
| PropEn mix xy2xy | $100.00 \pm 0.00$ | $100.00 \pm 0.00$ | $100.00 \pm 0.00$ | $100.00 \pm 0.00$ | $100.00 \pm 0.00$ | $100.00 \pm 0.00$ |
| PropEn x2x | $100.00 \pm 0.00$ | $100.00 \pm 0.00$ | $100.00 \pm 0.00$ | $100.00 \pm 0.00$ | $100.00 \pm 0.00$ | $100.00 \pm 0.00$ |
| PropEn xy2xy | $100.00 \pm 0.00$ | $100.00 \pm 0.00$ | $100.00 \pm 0.00$ | $100.00 \pm 0.00$ | $100.00 \pm 0.00$ | $100.00 \pm 0.00$ |
| Explicit | $100.00 \pm 0.00$ | $100.00 \pm 0.00$ | $100.00 \pm 0.00$ | $100.00 \pm 0.00$ | $100.00 \pm 0.00$ | $100.00 \pm 0.00$ |

| | N=50 | | | N=100 | | |
|---|---|---|---|---|---|---|
| | d=10 | d=50 | d=100 | d=10 | d=50 | d=100 |

**8-Gaussians**

**ratio imp**

| | N=50 d=10 | d=50 | d=100 | N=100 d=10 | d=50 | d=100 |
|---|---|---|---|---|---|---|
| **PropEn mix x2x** | $82.71 \pm 11.66$ | $80.53 \pm 9.30$ | $82.24 \pm 7.41$ | $92.56 \pm 4.27$ | $94.02 \pm 4.05$ | $94.02 \pm 6.21$ |
| **PropEn mix xy2xy** | $70.00 \pm 14.05$ | $82.71 \pm 8.44$ | $81.58 \pm 5.52$ | $90.38 \pm 6.09$ | $82.62 \pm 9.57$ | $89.74 \pm 7.55$ |
| **PropEn x2x** | $81.87 \pm 12.65$ | $85.96 \pm 10.85$ | $78.95 \pm 8.15$ | $92.82 \pm 3.78$ | $93.08 \pm 6.74$ | $93.41 \pm 5.71$ |
| **PropEn xy2xy** | $66.45 \pm 11.23$ | $78.20 \pm 8.28$ | $78.95 \pm 13.42$ | $88.28 \pm 5.31$ | $80.22 \pm 7.79$ | $82.05 \pm 9.85$ |
| **Explicit** | $49.28 \pm 10.86$ | $51.20 \pm 6.27$ | $48.80 \pm 6.70$ | $50.12 \pm 6.42$ | $48.02 \pm 4.30$ | $50.12 \pm 3.69$ |

**loglike**

| | | | | | | |
|---|---|---|---|---|---|---|
| **PropEn mix x2x** | $-42.61 \pm 3.91$ | $-41.18 \pm 2.06$ | $-41.42 \pm 1.99$ | $-87.75 \pm 2.80$ | $-87.99 \pm 3.89$ | $-85.62 \pm 3.12$ |
| **PropEn mix xy2xy** | $-43.81 \pm 3.17$ | $-42.28 \pm 2.05$ | $-42.15 \pm 2.37$ | $-87.73 \pm 3.09$ | $-89.18 \pm 3.40$ | $-88.79 \pm 4.11$ |
| **PropEn x2x** | $-40.63 \pm 3.69$ | $-39.08 \pm 2.96$ | $-39.54 \pm 0.79$ | $-80.22 \pm 3.54$ | $-79.61 \pm 4.29$ | $-80.38 \pm 3.74$ |
| **PropEn xy2xy** | $-43.62 \pm 4.15$ | $-40.37 \pm 2.51$ | $-40.84 \pm 4.53$ | $-82.36 \pm 3.51$ | $-84.59 \pm 6.19$ | $-84.04 \pm 4.75$ |
| **Explicit** | $-45.00 \pm 2.50$ | $-44.11 \pm 2.45$ | $-44.64 \pm 1.71$ | $-99.35 \pm 3.01$ | $-99.63 \pm 3.84$ | $-98.90 \pm 3.63$ |

**uniqness**

| | | | | | | |
|---|---|---|---|---|---|---|
| **PropEn mix x2x** | $100.00 \pm 0.00$ | $100.00 \pm 0.00$ | $100.00 \pm 0.00$ | $100.00 \pm 0.00$ | $100.00 \pm 0.00$ | $100.00 \pm 0.00$ |
| **PropEn mix xy2xy** | $100.00 \pm 0.00$ | $100.00 \pm 0.00$ | $100.00 \pm 0.00$ | $100.00 \pm 0.00$ | $100.00 \pm 0.00$ | $100.00 \pm 0.00$ |
| **PropEn x2x** | $100.00 \pm 0.00$ | $100.00 \pm 0.00$ | $100.00 \pm 0.00$ | $100.00 \pm 0.00$ | $100.00 \pm 0.00$ | $100.00 \pm 0.00$ |
| **PropEn xy2xy** | $100.00 \pm 0.00$ | $100.00 \pm 0.00$ | $100.00 \pm 0.00$ | $100.00 \pm 0.00$ | $100.00 \pm 0.00$ | $100.00 \pm 0.00$ |
| **Explicit** | $100.00 \pm 0.00$ | $100.00 \pm 0.00$ | $100.00 \pm 0.00$ | $100.00 \pm 0.00$ | $100.00 \pm 0.00$ | $100.00 \pm 0.00$ |

**novelty**

| | | | | | | |
|---|---|---|---|---|---|---|
| **PropEn mix x2x** | $100.00 \pm 0.00$ | $100.00 \pm 0.00$ | $100.00 \pm 0.00$ | $100.00 \pm 0.00$ | $100.00 \pm 0.00$ | $100.00 \pm 0.00$ |
| **PropEn mix xy2xy** | $100.00 \pm 0.00$ | $100.00 \pm 0.00$ | $100.00 \pm 0.00$ | $100.00 \pm 0.00$ | $100.00 \pm 0.00$ | $100.00 \pm 0.00$ |
| **PropEn x2x** | $100.00 \pm 0.00$ | $100.00 \pm 0.00$ | $100.00 \pm 0.00$ | $100.00 \pm 0.00$ | $100.00 \pm 0.00$ | $100.00 \pm 0.00$ |
| **PropEn xy2xy** | $100.00 \pm 0.00$ | $100.00 \pm 0.00$ | $100.00 \pm 0.00$ | $100.00 \pm 0.00$ | $100.00 \pm 0.00$ | $100.00 \pm 0.00$ |
| **Explicit** | $100.00 \pm 0.00$ | $100.00 \pm 0.00$ | $100.00 \pm 0.00$ | $100.00 \pm 0.00$ | $100.00 \pm 0.00$ | $100.00 \pm 0.00$ |

**pinwheel**

**ratio imp**

| | | | | | | |
|---|---|---|---|---|---|---|
| **PropEn mix x2x** | $76.13 \pm 8.52$ | $74.69 \pm 9.07$ | $70.18 \pm 14.31$ | $84.98 \pm 4.55$ | $85.58 \pm 8.33$ | $81.79 \pm 7.09$ |
| **PropEn mix xy2xy** | $63.52 \pm 8.45$ | $77.89 \pm 6.86$ | $70.39 \pm 14.60$ | $70.70 \pm 8.74$ | $63.53 \pm 12.08$ | $76.15 \pm 8.02$ |
| **PropEn x2x** | $80.77 \pm 9.76$ | $77.78 \pm 7.34$ | $84.21 \pm 9.85$ | $86.15 \pm 5.57$ | $82.05 \pm 5.73$ | $84.05 \pm 7.00$ |
| **PropEn xy2xy** | $61.79 \pm 5.53$ | $60.53 \pm 14.07$ | $67.42 \pm 13.04$ | $61.54 \pm 14.35$ | $50.43 \pm 7.01$ | $57.81 \pm 14.70$ |
| **Explicit** | $51.46 \pm 11.62$ | $46.62 \pm 8.34$ | $49.04 \pm 7.47$ | $48.48 \pm 4.65$ | $49.65 \pm 4.33$ | $50.58 \pm 3.82$ |

**loglike**

| | | | | | | |
|---|---|---|---|---|---|---|
| **PropEn mix x2x** | $-38.12 \pm 2.63$ | $-38.42 \pm 3.01$ | $-39.73 \pm 5.12$ | $-80.66 \pm 5.16$ | $-82.53 \pm 2.71$ | $-82.38 \pm 5.54$ |
| **PropEn mix xy2xy** | $-39.38 \pm 2.92$ | $-37.39 \pm 2.91$ | $-39.10 \pm 3.74$ | $-87.02 \pm 8.20$ | $-86.08 \pm 5.19$ | $-84.10 \pm 6.25$ |
| **PropEn x2x** | $-37.56 \pm 1.90$ | $-37.00 \pm 4.90$ | $-36.36 \pm 6.56$ | $-80.00 \pm 6.26$ | $-76.48 \pm 2.88$ | $-76.90 \pm 3.46$ |
| **PropEn xy2xy** | $-40.92 \pm 3.45$ | $-41.60 \pm 3.88$ | $-39.22 \pm 4.34$ | $-86.67 \pm 6.00$ | $-89.45 \pm 8.21$ | $-91.71 \pm 13.46$ |
| **Explicit** | $-40.40 \pm 1.93$ | $-39.58 \pm 2.24$ | $-40.15 \pm 1.81$ | $-90.35 \pm 5.03$ | $-88.91 \pm 3.16$ | $-87.91 \pm 4.62$ |

**uniqness**

| | | | | | | |
|---|---|---|---|---|---|---|
| **PropEn mix x2x** | $100.00 \pm 0.00$ | $100.00 \pm 0.00$ | $100.00 \pm 0.00$ | $100.00 \pm 0.00$ | $100.00 \pm 0.00$ | $100.00 \pm 0.00$ |
| **PropEn mix xy2xy** | $100.00 \pm 0.00$ | $100.00 \pm 0.00$ | $100.00 \pm 0.00$ | $100.00 \pm 0.00$ | $100.00 \pm 0.00$ | $100.00 \pm 0.00$ |
| **PropEn x2x** | $100.00 \pm 0.00$ | $100.00 \pm 0.00$ | $100.00 \pm 0.00$ | $99.23 \pm 2.43$ | $100.00 \pm 0.00$ | $100.00 \pm 0.00$ |
| **PropEn xy2xy** | $100.00 \pm 0.00$ | $100.00 \pm 0.00$ | $100.00 \pm 0.00$ | $100.00 \pm 0.00$ | $100.00 \pm 0.00$ | $100.00 \pm 0.00$ |
| **Explicit** | $100.00 \pm 0.00$ | $100.00 \pm 0.00$ | $100.00 \pm 0.00$ | $100.00 \pm 0.00$ | $100.00 \pm 0.00$ | $100.00 \pm 0.00$ |

**novelty**

| | | | | | | |
|---|---|---|---|---|---|---|
| **PropEn mix x2x** | $100.00 \pm 0.00$ | $100.00 \pm 0.00$ | $100.00 \pm 0.00$ | $100.00 \pm 0.00$ | $100.00 \pm 0.00$ | $100.00 \pm 0.00$ |
| **PropEn mix xy2xy** | $100.00 \pm 0.00$ | $100.00 \pm 0.00$ | $100.00 \pm 0.00$ | $100.00 \pm 0.00$ | $100.00 \pm 0.00$ | $100.00 \pm 0.00$ |
| **PropEn x2x** | $100.00 \pm 0.00$ | $100.00 \pm 0.00$ | $100.00 \pm 0.00$ | $100.00 \pm 0.00$ | $100.00 \pm 0.00$ | $100.00 \pm 0.00$ |
| **PropEn xy2xy** | $100.00 \pm 0.00$ | $100.00 \pm 0.00$ | $100.00 \pm 0.00$ | $100.00 \pm 0.00$ | $100.00 \pm 0.00$ | $100.00 \pm 0.00$ |
| **Explicit** | $100.00 \pm 0.00$ | $100.00 \pm 0.00$ | $100.00 \pm 0.00$ | $100.00 \pm 0.00$ | $100.00 \pm 0.00$ | $100.00 \pm 0.00$ |

