# OpenReview forum: "Implicitly Guided Design with PropEn: Match your Data to Follow the Gradient"
_NeurIPS.cc/2024/Conference — NeurIPS 2024 poster_

### Official Review · Reviewer_KGXt · 2024-07-10

**Soundness:** 3
**Presentation:** 3
**Contribution:** 3
**Rating:** 6
**Confidence:** 4

**Summary:**

The paper develops a simple but novel deep learning framework to implicitly optimize a seed input design to iteratively improve upon a particular property $$ g(\cdot) $$, until it reaches a fixed-point. In this way, the paper develops a novel framework for design optimization and provides a theoretical derivation to showcase that this implicit setup leads to an optimal learned neural network function $$ f^*(\cdot) $$ whose iterative refinement of a seed design , approximates the direction of gradient of the property function of interest. The framework has been evaluated on three diverse design tasks and compared with several state-of-the-art design methods.

**Strengths:**

- The proposed method is well motivated and novel for the problem of design optimization with neural networks without adversarial classification (which are inherently hard to train).


- The various experiments on toy as well as realistic design tasks like airfoil design optimization, antibody binding affinity design optimization demonstrate that the proposed method based on a simple `pair-wise matching` criterion is able to achieve better results than state-of-the-art models previously used for the same design task.


- The insight and the proof demonstrating that the constructed optimal function $f^*(\cdot)$ approximates the direction of the gradient of a property function $g(\cdot) $ even though it is not explicitly trained to do so, is interesting and will benefit the space of design optimization.

**Weaknesses:**

- Further analysis about the effect of $ \Delta_x $ and $\Delta_y  $, their selection methodology and their sensitivity in various design contexts is necessary as the matching step critically depends on these two parameters.


- The method has been demonstrated only in the context of designs that improve a single property $ g(\cdot) $. However in reality designs usually need to consider multiple properties simultaneously. This brings into question the applicability of the current method to practical problems.


- A rigorous analysis of the "goodness" of the seed design and the effect of said goodness on the convergence speed / accuracy is necessary.

**Questions:**

1. How are $ \Delta_x $ and $\Delta_y  $ set / tuned?

2. How does property under consideration $ g(\cdot) $ influence the calibration / selection of $ \Delta_y $?

3. How can the current model be augmented to function with multiple properties that might need to be simultaneously improved?

4. How are seed designs selected for initial inputs? i.e, what is considered a minimum viable seed design to be improved upon / refined until convergence? How does the model react to bad (i.e., flawed / noisy) designs?

**Limitations:**

A limitation of the current work is its inability to incorporate more than a single property of interest. However, the authors have explicitly mentioned this limitation and the reviewer believes that despite this, the proposed method is still valuable and can serve as a good foundation for future research in design optimization.

---

> ### Author Rebuttal · Authors · 2024-08-07
>
> **[Setting $\Delta_x$ and $\Delta_y$]**
>
> The choice of parameters $\Delta_x$ and $\Delta_y$ should be informed by the specific application. For example, in antibody design, domain experts recommend not using thresholds above a Levenshtein distance of 8, as such differences are considered biologically irrelevant. Similarly, for $\Delta_y$, knowing that noise in binding measurements can be up to 0.3, we chose 0.5 to ensure proper matching.
>
> When faced with a new dataset of \(n\) unique data points, a good initial approach is to use the standard deviation of pairwise distances for \(x\), and the mean or median for property \(y\):
>
> - $
> \Delta_x \lt \sigma_d = \sqrt{\frac{2}{n(n-1)} \sum_{1 \leq i < j \leq n} \left( d(x_i, x_j) - \frac{2}{n(n-1)} \sum_{1 \leq k < l \leq n} d(x_k, x_l) \right)^2}
> $
>
> - $
> \Delta_y \lt \frac{2}{n(n-1)} \sum_{1 \leq i < j \leq n} d(x_i, x_j)
> $
>
>
> **[How g(.) influences the calibration/selection of d_y]**
> When the $g(.)$ is non-smooth, or we have a very few datapoints, this certainly influences the choice of $\Delta_y$, in this case we opt for as small steps as possible since this will create many pairs which should give sense of the direction of the gradient around the sparse regions.
> In the attached PDF, we provide an additional ablation study on how threshold choices affect the number of training pairs, showing that sufficiently large thresholds include all unique training points. For performance impact, please refer to the ablation study in Figure 4.
>
> **[Multiple-Property Optimization]**
>
> We agree that multiple-property optimization is crucial in real-world applications, as we have experienced ourselves. In such scenarios, we compute a multivariate rank over all properties of interest and optimize for that single score with PropEn. The multivariate score can be computed using the Hypervolume [1] or the CDF indicator, as suggested by [2].  Both methods work effectively, and we provide results on in-silico multi-property antibody optimization in the attached PDF. We select four developability properties: number of liabilities, hyrophobicity, positive and negative charge. We compute a multi-property ranking over these 4 properties, resulting in a single, scalar score we will use in PropEn. We then train PropEn models to optimize for each of these properties individually, and an *mPropEn* which uses the multivariate ranking score. In the figure in the attached pdf, we observe that (1) for two orthogonal targets, mPropEn suggests designs on the Pareto Front, while the single-objective properties tend to suggest designs at the tails of the front as expected, and (2) taking all four objectives into account, the top 20% of designs at the Pareto front come from the variants of mPropEn. This validates our idea of combining PropEn with multi-objective indicators, and we plan to expand on other datasets and experiments in further analysis.
>
> [1]
> Zitzler, Eckart, et al. "Performance assessment of multiobjective optimizers: An analysis and review." IEEE Transactions on evolutionary computation 7.2 (2003): 117-132.
>
> [2] Park, Ji Won, et al. "BOtied: Multi-objective Bayesian optimization with tied multivariate ranks."ICML 2024.
>
> **[Seed Designs Selection and Flawed Designs]**
>
> Generally, seed designs come from the training data. In cases where they do not, our model's tendency to optimize locally ensures that out-of-distribution (OOD) seeds are mapped to the closest neighborhood in the training set, as shown in Figure 2 in the submission and figure 1 and 2 in the attached pdf. In contrast, other methods, such as explicit guidance, often fall off the manifold and suggest designs in regions unsupported by the training data.
>
> We have found that starting with seeds at the distribution's edge, with the highest possible values, helps achieve significant improvements, as these seeds have the best chance of extrapolating. The attached PDF provides examples of seeds where PropEn achieved higher values than anything observed in the training distribution.

---

> > ### Author Response · Authors · 2024-08-12
> > **Official Comment by Authors**
> >
> > Dear Reviewer,
> >
> > Thank you for your valuable feedback and support. As the author-reviewer discussion period deadline approaches (Aug 13), we kindly remind you to provide any additional feedback. We hope our responses have addressed your concerns, and if so, would greatly appreciate a score update. If there are remaining issues, please let us know, and we will promptly address them.
> >
> > Best regards.

---

### Official Review · Reviewer_gEZM · 2024-07-13

**Soundness:** 3
**Presentation:** 3
**Contribution:** 2
**Rating:** 6
**Confidence:** 4

**Summary:**

The paper address design optimization, which is the process of optimizing over a "design" parameter space to optimize over one or more observable outcomes in many scientific and engineering problems. The proposed framework "PropEn" uses a three step process, first identifying a "matched dataset" that pairs every sample with another with a better outcome; this is followed by training a model that takes a sample and predicts, essentially creating an autoregressive sampler. The technique is evaluated on multiple science and engineering benchmarks.

**Strengths:**

* The matched objective is an interesting application of regressive sampling, and a different view on augmenting a small dataset, which is a common issue in many science problems.
* The paper goes into theoretical justifications as to why a matched reconstruction objective is useful, suggesting that it approximates the gradient of an explicitly trained property predictor.
* Its also shown that the generated samples have a high likelihood close to the training samples, indicating that they are indeed sampling from the training distribution.

**Weaknesses:**

* PropEn is an interesting idea for design optimization but it needs to be more thoroughly benchmarked against existing methods. For example, there is a host of methods that do Bayesian optimization, and variants thereof, that can work with as few as 10 samples (albeit in lower dimensions) which the paper does not consider. I think it sheds light into the behavior of PropEn in lower dimensions. More recent methods that use DNN based UQ estimates are able to go beyond 5-10D datasets effectively as well.
* At a high level, matching and matched reconstruction appear to be very closely related to diffusion models. In my understanding, the diffusion process can be framed as matched reconstruction, where the property to maximize is the likelihood of the sample. Framing matched reconstruction through the lens of diffusion might not only strengthen the paper's formulation but also provide other theoretical insights.
* Its unclear to me why neighborhoods in the design space are constructed using L2-balls, why is this the best way to identify similarity in x? How sensitive is the optimization to the size of the neighborhood? Or in another sense, a related question is -- how sensitive is matched reconstruction to the choice of sampling? for e.g. if you start with an LHC sampling (or any other variants) vs other types.
* On small amounts of data, what kinds of regularizations keep the main model from overfitting?

**Questions:**

Please see above

**Limitations:**

Yes, it appears so.

---

> ### Author Rebuttal · Authors · 2024-08-07
>
> **[Benchmarking Against Bayesian Optimization (BO)]**
> Please see our general response.
>
> **[Connection to Diffusion Models]**
> Please see our general response.
>
> **[Neighborhoods in L2 Ball]**
>
> The assumption regarding neighborhoods in L2 balls was made solely for theoretical purposes, to make the explanation more general and accessible. Neighborhoods in L2 balls naturally connect to training with Mean Squared Error (MSE) loss. However, the methodology is versatile and can be adapted to train with different losses; it can be shown that the same theory holds for Binary Cross-Entropy (BCE) loss as well. We are happy to expand on the proof in further discussion if the reviewer finds this interesting.
>
> Similarity in \(x\) should be defined based on the application, which is why we showcase different datasets. For antibodies, we used Levenshtein distance as an example. We believe this choice gives freedom to include more inductive biases from domain experts.
>
> **[Sensitivity of Matched Reconstruction to Sampling Choice]**
>
> There is no sampling choice made in PropEn; we keep all the possible pairs from the training data. This approach ensures that our method is comprehensive and does not rely on arbitrary sampling decisions.
>
> **[Regularization to Prevent Overfitting]**
>
> We have identified two ways to avoid overfitting:
>
> 1. Mixup technique: Including a reconstruction loss to the original sample in the training loss (the mixup variant in experiments) helps prevent overfitting. Due to the one-to-many matching scheme, we cannot perfectly reconstruct a single example, but rather learn how to interpolate between samples.
>
> 2. Property Value Inclusion: Incorporating the value of the property in reconstruction controls the gradient steps (the xy2xy variant).
>
> The theoretical background on why this works is included in the supplementary material, Section A.4, "Understanding the Matched Reconstruction Objective."

---

> > ### Author Response · Authors · 2024-08-12
> > **Official Comment by Authors**
> >
> > Dear Reviewer,
> >
> > Thank you for your valuable feedback and support. As the author-reviewer discussion period deadline approaches (Aug 13), we kindly remind you to provide any additional feedback. We hope our responses have addressed your concerns, and if so, would greatly appreciate a score update. If there are remaining issues, please let us know, and we will promptly address them.
> >
> > Best regards.

---

> > ### Comment · Reviewer_gEZM · 2024-08-12
> >
> > Thank you for the response and overall comments. I do appreciate the authors on actually implementing the designs into a wet lab framework, as a test for real world applicability -- a high bar for evaluating the impact of the work. A note on BO/AL, i think the distinction the rebuttal makes is subtle and it would still be useful to compare against BO/AL on known benchmarks to show the benefits, considering the broad applicability of BO/AL methods. There have been works in the past that use BO in conjunction with a generative models, this may be an informative experiment. In light of these comments, I will raise my score to 6.

---

> > > ### Author Response · Authors · 2024-08-14
> > >
> > > Thank you for reconsidering your score recommendation. In addition to the comparison with Lambo, which uses BO to guide design optimization, we could include an experiment in the supplementary of the final manuscript that evaluates PropEn in BO on standard test functions like Branin-Currin or DTLZ, where we have ground truth solutions for comparison. This would involve a different problem setup, considering multiple rounds, and would require adapting PropEn to handle design selection independently. We will also include an extended discussion on this point.
> > >
> > > Best regards

---

### Official Review · Reviewer_zYK4 · 2024-07-19

**Soundness:** 2
**Presentation:** 3
**Contribution:** 2
**Rating:** 5
**Confidence:** 3

**Summary:**

This work proposes the method PropEn, which is inspired by the concept of matching techniques in econometrics. Using PropEn (specifically in scenarios with a lack of large datasets), the authors can expand the dataset, which will inherently help in design improvement, etc. To do this, they train a network to learn a mapping from an initial sample to another sample with an improved target attribute selected during the dataset curation phase. This model can further be sampled auto-regressively until it converges to a final candidate design.

**Strengths:**

- The paper is very well written, and the motivation, methods, and results are clearly presented and easy to understand.
- The proposed method is simple, generalizable, and appears effective.
- The experiments and corresponding results are well-defined and support the claims made in the paper.

**Weaknesses:**

- Additional experiments that would add substantial value to the work would be to extend the data matching method to other models, rather than a simple encoder-decoder model with a reconstruction loss. As an example, provide experiments for diffusion models where the conditioning signal is the x_0 seed (or x_i in autoregressive sampling), and the generated design is x_{i+1}, which would solidify your claims about the generalizability of the data matching method.
- Have the authors considered active learning approaches? This work is very close to active learning ideas, applied in the context of low data regimes.
- The aspect of evaluating the target properties on the neighboring samples is not clearly explained.
- the examples shown are simple and not really huge.

**Questions:**

See above in weakness

**Limitations:**

No limitations are mentioned by the authors.

---

> ### Author Rebuttal · Authors · 2024-08-07
>
> **[Extending Matching to Other Models]**
> Please see our general response.
>
> **[Comparison to Active Learning]**
> Please see our general response.
>
> **[Evaluating Target Properties on Neighbouring Samples**]
>
> Could you please elaborate on what you mean by "evaluating the properties on neighbouring samples"? We are not entirely sure about the question, but we hope this explanation helps: The matching is based on observed or measured values of properties for each example; we do not use any predictor to obtain them. For the toy example, the property was computed using a KDE estimator. For the airfoils, lift and drag coefficients were obtained by Neural Foil, and for the antibodies, binding affinity was measured through wet lab experiments.
>
> **[Complexity and Diversity of Examples]**
>
> We made an effort to demonstrate that this method works across different domains, neural network architectures, and tasks. Typically, machine learning papers follow a standard benchmark within a single application domain. Our experiments cover toy, engineering, and biology applications, showcasing a broader diversity. Additionally, we include results from wet-lab experiments, which cost $20,000 and take several months to complete. We hope the reviewer will reconsider and examine our manuscript with attention to these aspects.

---

> ### Comment · Reviewer_zYK4 · 2024-08-11
>
> I have read the comments and am satisfied with the clarifications and additional results. I will raise my score to borderline accept

---

> > ### Author Response · Authors · 2024-08-12
> > **Official Comment by Authors**
> >
> > Dear Reviewer,
> >
> > Thank you for taking the time to provide your valuable feedback. We are glad to hear that the clarifications and additional results we provided addressed your concerns and we are grateful for your decision to raise the score.
> > Best regards.

---

### Official Review · Reviewer_WPJM · 2024-07-23

**Soundness:** 2
**Presentation:** 3
**Contribution:** 3
**Rating:** 6
**Confidence:** 2

**Summary:**

The paper presents a generative framework for property enhancement. The proposed framework consists of only a generative model, and it's missing a discriminator that is usually found in other frameworks for guided design. This is achieved by training the generative model on a "matched" dataset that consists of paired examples $(x, x')$ where $x'$ is an "enhanced" version of $x$ from its immediate proximity ("enhanced" meaning it has a higher value of the property that is sought to be maximized in the guided design process, and "proximity" depends on the problem domain).

A mathematical analysis of the proposed method shows that the learnt generative function converges to the gradient of the function computing the property and that examples sampled by the function are as likely as the training set. These analysis motivate the iterative application of the generative function to converge to a stationary point with enhanced property.

The authors claim that this framework is particularly well suited for applications with scarce data, and apply it to a toy problem and airfoil optimization, as well as protein optimization. In the latter application the method shows superior performance compared to other methods, as measured by wetlab experiments.

**Strengths:**

1. The paper presents a simple method with mathematical justifications why it should work.

1. Additionally to the toy examples, the method is applied to airfoil enhancement and protein optimization. The predictions of the latter are confirmed by wetlab experiments, and the reported results are better than state-of-the art methods.

**Weaknesses:**

1. The paper is missing several important implementation details. I would be especially interested in knowing the details of the training datasets and the constructed matched datasets, which I could not find anywhere in the paper or the appendix. Section B.1 in the appendix should be expanded to explain in detail what models were used. For example, line 571 states that a "ResNet with 3 blocks" was used to generate one-hot encoded AhO aligned sequences. But it's not clear to my why a 2D convolutional architecture would be used for such a problem, how exactly the data was represented, and what motivates the use of that precise model.

1. The airfoil optimization is missing a strong baseline. From the presented results I find it hard to judge whether the method indeed works well across domains. I'm not familiar with the protein optimization experiments, so I cannot judge the power of the method by the presented results, especially since important implementation details are missing (see above weakness). Since all the findings in Section 3.2 are based on very few datapoints (per optimization target) and there is a wide intra-method variability, I would like to see some statistical analysis for presented data.

**Questions:**

1. Section 2.3 lines 126-127 "As a property to optimize, we choose the log-likelihood of the data as estimated by a KDE with Gaussian kernel" – it seems that optimizing for likely examples would both optimize the property as well as the likelihood of the examples. But theorem 1 is about increasing the property value and theorem 2 is about sampling likely examples. So I would have thought it more natural to choose an illustrative example that decouples the property to be optimized from the likelihood of the examples.

1. I couldn't find the numbers reported in Table 3 in referenced papers [38, 39]. If it is a reproduction by the authors, then I would like to see that clearly stated in the main text or the table caption.

1. The assumption of a "sufficiently overparametrized model" (line 472) seems a pretty strong one to me. How realistic is this assumption in the presented settings, and what would be the consequence for Theorem 1, and PropEn applicability in general, if the assumption does not hold?

Various nits, questions, comments:
1. line 33: it should say "even for deep neural networks" (remove "with")
1. caption Figure: missing space "value.Bottom"
1. line 39: missing a word at "[6] the"
1. line 48: "have only be used provide"
1. line 57: "PropEn", not "Propen"
1. line 113: "from the its"
1. line 125: please add a reference to the "well-konwn 2d pinwheel dataset" (I don't know which dataset is meant exactly); similarly, the "8-Gaussian" dataset (line 157) is also missing a reference
1. line 154: why is the regularized variant called "mixup/mix"? I find this confusing, as there is a well-known mixup technique in ML that does not seem related (https://arxiv.org/abs/1710.09412)
1. line 165: how are metrics "uniqueness" and "novelty" defined?
1. line 185: reference for "NACA airfoils" missing
1. Section 3.1.2: how were the initial airfoil designs chosen?
1. lines 216-218 and Figure 4: when varying one threshold, what is the value of the other threshold? a 2D density plot might point a more complete picture of the measured improvement when varying both thresholds.
1. lines 219-222 and Figure 4: we can see that in a single example the $C_l/C_d$ ratio increased monotonically; but it's quite a stretch from that to claim that "all designs ... are deemed plausible" and that there is (always) a "consistent enhancement"; Figure 7 in the appendix (which is not mentioned in the main paper) does not add substantially to these claims
1. line 225: it would be nice to reference figures/tables to show that "mixup variant of PropEn may require longer training"
1. line 234: "referred", not "refereed"
1. Figure 5 (left): why do the points have different sizes?
1. Figure 8: what is the take-away from this figure?

**Limitations:**

The "NeurIPS Paper Checklist" in the appendix still needs to be filled in (currently it only shows the instructions).

---

> ### Author Rebuttal · Authors · 2024-08-07
>
> **[Details on Training and Matched Datasets]**
> We add the following table in the supplement:
>
> | Dataset         | # Pairs | # Unique Samples in Train | Y Range (Control) | Y Range (Treatment) |
> |-----------------|---------|---------------------------|-------------------|---------------------|
> | 8 Gaussians     | 1,746   | 96                        | [0.14, 0.86]      | [0, 1]              |
> | Airfoils        | 8,125   | 200                       | [56.3, 82.8]      | [65.2, 90.6]        |
> | Antibodies (T1) | 1,362   | 268                       | [4, 8.9]          | [6.5, 9.3]          |
>
>
> **[Details on Models and the choice of ResNet]**
>
> We missed mentioning that the ResNet blocks in our network are non-convolutional. PropEn for antibodies uses fully connected layers with residual connections, one-hot encoding for input sequences, and layer normalization and GELU activation. This choice is tailored to the specific demands of antibody sequence analysis.
>
> **[Airfoil Optimization Baseline]**
>
> We added a diffusion baseline for the airfoil experiment following the implementation from [1]. The model is a 3-layer MLP with 128 units per layer, sinusoidal time and input embeddings, and GELU activations. We conducted a grid search over batch size, hidden layer size, learning rate, and time steps and repeated the experiment five times for the best parameters.
> Results for $T \in \{ 5, 15, 50\}\$ show that the diffusion model optimizes almost all hold-out designs, however with minor improvements in the lift-drag ratio. For the final manuscript, we'll explore more advanced diffusion models, particularly from reference [2].
>
> 1. [tiny-diffusion](https://github.com/tanelp/tiny-diffusion)
> 2. [Diffusion-based 2DAirfoil Generation](https://github.com/tonyzyl/Diffusion-based-2DAirfoil-Generation/blob/main/models/airfoil_MLP.py)
>
> **[Statistical Analysis for Results in Section 3.2]**
>
> The expense of wet-lab experiments constrains us, limiting the number of tested examples. The overall cost for the results presented was approximately $20,000. State-of-the-art methods like Walk-Jump and Lambo include results from a single target and seed, whereas we expand to 4 targets and 8 seeds.
>
> As requested, we add statistical significance tests using Fisher's Exact test and Chi-Square tests.
>
> **Statistical Significance Tests for Table 3: Binding rate per method;**
>
> | Propen vs               | Fisher Odds Ratio | Fisher P-Val | Chi-Square stst| Chi-Square P-Val |
> |--------------------|-------------------|----------------|----------------------|---------------------|
> | Walk-Jump          | 10.4000           | **0.0**        | 26.9855              | **0.0**              |
> | Lambo (Guided)     | 22.2316           | **0.00**         | 41.6322              | **0.0**              |
> | Diffusion          | 2.7077            | 0.1022         | 1.6451               | 0.1996              |
> | Diffusion (Guided) | 1.3538            | 0.4295         | 0.0304               | 0.8616              |
>
>
>
> **Statistical Significance Tests for Table 4: Number of designs improving over seed per method;**
>
> | PropEn vs       | Fisher Odds Ratio | Fisher P-Val | Chi-Square stat | Chi-Square P-Val|
> |--------------------|-------------------|----------------|----------------------|---------------------|
> | Walk-Jump          | 10.4918           |  **0.0**        | 26.6097              | **0.0**             |
> | Lambo (Guided)     | 3.2350            | **0.0098**         | 5.1369               | **0.0234**              |
> | Diffusion          | 2.8478            | **0.0156**         | 4.4274               | **0.0354**             |
> | Diffusion (Guided) | 1.7947            | 0.0586         | 2.4436               | 0.1180              |
>
>
> **[Toy Exp;  choice of property]**
>
> We propose a setup where the property and likelihood are disentangled, retaining the shape of 8 Gaussians but computing a property increasing in value in an anticlock-wise direction. Please see the attached PDF for results.
>
> **[Numbers in Table 3]**
>
> We only reference the source publications and we independently run each of them our wet-lab experiments. This is why the numbers do not appear in their corresponding publications.
>
> **[Overparameterized Model Assumption]**
> This is a general, common assumption in deep learning models. We will add a discussion on the overparameterized model; essentially, it is a standard assumption that has been theoretically [1, 2] and empirically explored in many prior works on deep learning. Overparameterization refers to the practice of using models with more parameters than the number of data points or the complexity strictly required to fit the training data. This assumption might seem counterintuitive initially, as it suggests using models that are larger than necessary. However, several compelling arguments justify this approach, highlighting its benefits for model performance, generalization, and learning dynamics.
>
> [1] Bengio, Y. et al. "Representation learning: A review and new perspectives." IEEE transactions on pattern analysis and machine intelligence".
> [2] Allen-Zhu et al  "A convergence theory for deep learning via over-parameterization." ICML 2019.
>
> **[Response to minor comments]**
>
> [Mixup/Mix]
> This version of PropEn, similar to the original Mixup you referenced, interpolates between examples. If this is confusing, we can consider renaming it.
>
> [Uniqueness and Novelty]
> - Uniqueness:  #unique designs divided by #gen designs.
> - Novelty: #designs proposed by the method that don't appear in the training data, divided by the #gen designs.
>
> [How Were the Initial Airfoil Designs Chosen?]
> Random holdout dataset.
>
> [Figure 4]
> When varying \(\Delta_x\), \(\Delta_y\) is set to 1, and vice versa.
>
> [Figure 5]
> The size corresponds to the number of designs for that seed.
>
> [Figure 8]
> Explicit guidance tends to fall off the manifold when many optimization steps are taken or seeds are OOD.

---

> > ### Author Response · Authors · 2024-08-12
> > **Official Comment by Authors**
> >
> > Dear Reviewer,
> >
> > Thank you for your valuable feedback and support. As the author-reviewer discussion period deadline approaches (Aug 13), we kindly remind you to provide any additional feedback. We hope our responses have addressed your concerns, and if so, would greatly appreciate a score update. If there are remaining issues, please let us know, and we will promptly address them.
> >
> > Best regards.

---

> > > ### Comment · Reviewer_WPJM · 2024-08-13
> > >
> > > Thank you for attentive replies to my original review.
> > >
> > > First some minor replies:
> > >
> > > [Mixup/Mix] Yes indeed, I still find this confusing: in the cited Mixup paper the regularization works by creating convex combinations from pairs of input data and labels. In the case of PropEn the regularized variant works by incorporating a reconstruction regularizer $ℓ(f_θ(x), x)$. It is not quite clear to me how this regularized PropEn variant "interpolates between examples", as you claim in your response.
> > >
> > > [Details on Models] Even with the additional piece of information that the ResNet is 1D I'm not quite sure about the exact models used. For example, the toy dataset model (Section B.1, lines 556-562) mentions a 2 layer MLP with 100 units per layer, and the main text mentions (Section 3.1.1, lines 162-163) that the explicit baseline uses an "auto-encoder of the same architecture as PropEn augmented with a discriminator for guidance". I would like to see more details about the discriminator and the encoder-decoder. More broadly, it would be nice to have a discussion about the models used and how they were selected. Especially since the authors choose their own baselines, care should be taken that not only the proposed method yields good results, but also that the baselines are properly optimized wrt their architecture and hyper parameters.
> > >
> > > [Concerning all comments in the response] Thank you for the clarifications, I assume you would add them also to the paper? (e.g. the response currently says "We add the following table in the supplement", but it doesn't mention whether the submission would be updated to any of the other replies).
> > >
> > > Then, some more comments on the main weakness raised in my initial comment:
> > >
> > > **[Missing comparison to strong baselines]** After reading the paper and the author's reply to my review, I'm still not convinced that the method really performs better than other available methods. The main problem in my opinion is that the authors implement their own baselines and the paper lacks details that convince that the methods are applied in such a way that they really yield best possible results.
> > >
> > > One way to mediate this would be to take strong results from existing papers and then show that PropEn performs better on the exact problems evaluated by these papers. Another way would be to show convincingly in the paper that the methods were tuned in such a way that they yield best possible results.
> > >
> > > As for the added Airfoil baseline, I'm not convinced that these are really strong baselines, as [Diffusion-based 2DAirfoil Generation](https://github.com/tonyzyl/Diffusion-based-2DAirfoil-Generation/blob/main/models/airfoil_MLP.py) is a Github repo with a single star, and [tiny-diffusion](https://github.com/tanelp/tiny-diffusion) is "A minimal PyTorch implementation of probabilistic diffusion models for 2D datasets", where Table 1 states that the Airfoil dataset has 400 dimensions.
> > >
> > > Overall, I appreciate the author's efforts in their reply, but I don't think that they sufficiently addressed the main weaknesses mentioned in my initial review, so I remain with my initial rating of 4 (borderline reject).

---

> > > > ### Author Response · Authors · 2024-08-13
> > > >
> > > > Thank you for reviewing our rebuttal and further clarifying your remarks.
> > > >
> > > > **[Mixup]** We will change the name of this variant to **Cross xx2xx**, and we are open to suggestions from the reviewer. By interpolation, we refer to the behaviour observed during training in this mode. Specifically, the network learns to reconstruct (a control sample) `x` into (a treatment sample) `x'`, `x''`, `x'''`, etc. This means the network never learns a perfect reconstruction of one specific treatment, but instead generates variants in-between the two.
> > > >
> > > > **[Details on Models]** The rebuttal response was limited to 6,000 characters, so we couldn't include all the details. Please note that in Section B.1, lines 556-562, we specify 30 neurons per layer, not 100 as mentioned in the reviewer's comment. To avoid confusion, please find below the networks used for the toy and engineering experiments, which we will add to the supplementary material of the paper. For WJS, we used a 1D Conv architecture (please see A.1 in [38]), and for Lambo, a bert-small transformer (please see C.1 in [39]).
> > > >
> > > > Please note that the code for all of the experiments will be released. For optimizing the parameters of the baselines in the toy and engineering experiments, we conducted a grid search over the learning rate (`[1e-2, 1e-5]`), weight decay (`[1e-2, 1e-5]`), number of epochs (`[300, 1000, 5000]`), batch size (`32, 64, 128`), and number of neurons per layer (`[30, 50, 100]`). For the antibody experiments, each baseline was implemented and trained as advised by the authors of the original publications.
> > > >
> > > > **Toy example architectures:**
> > > > AE(
> > > >   (criterion): MSELoss()
> > > >   (encoder): Sequential(
> > > >     (0): Linear(in_features=2, out_features=30, bias=True)
> > > >     (1): ReLU()
> > > >     (2): Linear(in_features=30, out_features=30, bias=True)
> > > >     (3): ReLU()
> > > >     (4): Linear(in_features=30, out_features=30, bias=True)
> > > >     (5): ReLU()
> > > >     (6): Linear(in_features=30, out_features=15, bias=True)
> > > >   )
> > > >   The decoder is analogous to the encoder, starts with the compressed representation (15) and progressively reconstructs the original input size (2 features).
> > > >
> > > > (Discriminator): Sequential(
> > > >     (0): Linear(in_features=2, out_features=30, bias=True)
> > > >     (1): ReLU()
> > > >     (2): Linear(in_features=30, out_features=30, bias=True)
> > > >     (3): ReLU()
> > > >     (4): Linear(in_features=30, out_features=30, bias=True)
> > > >     (5): ReLU()
> > > >     (6): Linear(in_features=30, out_features=1, bias=True)
> > > >   )
> > > > )
> > > >
> > > > Diffusion:
> > > >
> > > > MLP(
> > > >   (time_mlp): PositionalEmbedding(
> > > >     (layer): SinusoidalEmbedding()
> > > >   )
> > > >   (input_mlp1): PositionalEmbedding(
> > > >     (layer): SinusoidalEmbedding()
> > > >   )
> > > >   (input_mlp2): PositionalEmbedding(
> > > >     (layer): SinusoidalEmbedding()
> > > >   )
> > > >   (joint_mlp): Sequential(
> > > >     (0): Linear(in_features=2, out_features=128, bias=True)
> > > >     (1): GELU(approximate='none')
> > > >     (2): Block(
> > > >       (ff): Linear(in_features=128, out_features=128, bias=True)
> > > >       (act): GELU(approximate='none')
> > > >     )
> > > >     (3): Block(
> > > >       (ff): Linear(in_features=128, out_features=128, bias=True)
> > > >       (act): GELU(approximate='none')
> > > >     )
> > > >     (4): Block(
> > > >       (ff): Linear(in_features=128, out_features=128, bias=True)
> > > >       (act): GELU(approximate='none')
> > > >     (5): Linear(in_features=128, out_features=2, bias=True))
> > > >
> > > > **[Concerning all comments in the response]** Yes, we will add all modifications from the rebuttal to the manuscript. We appreciate all remarks and believe that by incorporating these modifications, we will improve our submission.
> > > >
> > > > **[Missing comparison to strong baselines]** We disagree with the reviewer regarding the lack of strong baselines:
> > > > * In the toy example, AE with guidance is the most standard baseline from the literature. As is common in all ML publications, to ensure a fair comparison, we compare the same architectures for the implicit and explicit baselines (after hyperparameter tuning).
> > > > * In the engineering example, tiny diffusion is the most appropriate baseline to compare to given the MLP architecture used for PropEn. The diffusion model was developed to generate hundreds of points in 2D, which is exactly the same as the airfoil setup (so the reviewer's concern about dimensionality mismatch and the inappropriateness of the baseline is not grounded). We believe this is the fairest baseline we could add for comparison.
> > > > * In the antibody design experiment, we compare against **three** SOTA methods.
> > > >
> > > > Finally, we would like to remind the reviewer that we never claim in the manuscript or the rebuttals that PropEn is the best method across the board. We believe the true advantage of PropEn is its **applicability and efficiency across many domains**, which is what we aim to confirm in our experimental evaluation. All other baselines are application-specific, requiring numerous modifications and inductive biases included in the architecture and training. This is also why it is difficult to have the same baselines across all applications, as most methods focus on a single application, while PropEn is general.

---

> > > > > ### Comment · Reviewer_WPJM · 2024-08-13
> > > > >
> > > > > Thank you for the quick and exhaustive reply to my latest comments.
> > > > >
> > > > > [Mixup] I think using any other name, such as "Cross xx2xx" would be clearer. Thank you for the update.
> > > > >
> > > > > [Details on Models] Thank you for the additional information about the used models. I think these details should also be briefly mentioned in the appendix.
> > > > >
> > > > > **[Missing comparisons to strong baselines]** Since I am not familiar with the guided design literature and airfoil design / protein optimization, I cannot get a solid impression of the performance of the proposed algorithm with the data presented in the paper. If the method were compared directly to results from the literature, that would certainly give a better grounded impression of the performance.
> > > > >
> > > > > But I broadly agree with the rebuttal from the authors, and since no other reviewer commented on this point, I have increased my rating accordingly.

---

> > > > > > ### Author Response · Authors · 2024-08-14
> > > > > >
> > > > > > We are pleased to hear that the reviewer agrees with our rebuttal, and we sincerely appreciate all the comments and remarks provided during the discussion period. We will incorporate the additional results and clarifications into our manuscript.
> > > > > >
> > > > > > Best regards

---

### Author Rebuttal · Authors · 2024-08-07

We would like to thank the reviewers for their time and thoughtful comments. Here we will summarize our response addressing common/essential concerns and then follow with point by point responses. In what follows we refer to the submited manuscript as ‘submission’ and the pdf accompanying our rebuttal as the ‘attached pdf’.

**[clarification]** We would first like to reiterate on the contributions of our work which we believe the reviewers might have missed or overlooked. We propose a general method, that is domain and architecture agnostic and can be straightforwardly applied to variety of tasks. In the submission we show experiments across various domains such as engineering and biology. We made effort to provide theoretical foundation for PropEn by relating it to implicitly approximating the gradient for the property of interest, and finally, we include an in-vitro validation experiment to compare to SOTA methods, that is, to the best of our knowledge, the most thorough experiment (see below) of this kind appearing in any publication (ML or otherwise).

**[value and significance of antibody optimization results]** We hope the reviewers and AC do see the value of including a diverse, expensive wet-lab experimetnts evaluation.
* It is very challenging, and therefore, exceedingly uncommon for machine learning papers to include in-vitro results -- much less ones that devote experimental budget to competing methods, that utilize expensive and highly accurate assays (SPR), and that include 3 targets and 8 seeds (previous works considered 1 seed). This is a contribution in itself because it shows the real applicability and advantage of PropEn.
* We are comparing PropEn head-to-head with exceptionally strong baselines: WalkJump received the best paper award in ICLR 2024 and Lambo was a spotlight in NeurIPS 2023.
* To strenghten our claims, we did a statistical significance test for the numbers in Table 3 and 4. Detailed results are included in our response to reviewer WPJM.

**[additional toy experiment]** as requested by reviewer WPJM we include a new toy example where the property is dissentangled from the likelihood of the data. We repeated the analysis as in the original submission, and all results can be found in the attached pdf. The conclusions are consistent with the disussion in the manuscript.

**[stronger baseline for the airfoil example]** We now add a diffusion baseline for the airfoil optimization application. The results in the attached pdf show that these models do improve the shape in many cases, however the improvement is not substantial. We hope this answers the concerns raised by revievewrs WPJM and zYK4.

**[multi-property optimization]** As requested by reviewer KGXt, we now include experiments on multi-property optimization by leveraging Hypervolume or CDF scores to compute and then optimize multivariate ranks. Our experiments show this strategy is effective and straightforward to use. Please see the attached pdf for results.

**[comparison to Bayesian Optimization/Active Learning (BO/AL)]**
This point was raised by two reviewers gEZM and zYK4.
While both BO and PropEn aim at optimization in small sample sizes, the two frameworks solve different problems. The goal of PropEn is to generate designs, whereas in BO/AL the goal is to choose the most promising designs that should be labeled in order to imporve a predictors performance or find the best candidate, i.e. the focus is selection. In the context of optimizing designs, one would have a suite of (1) generative models, (2) property predictors and (3) BO/AL module that will do the final selection across pool of candidates. PropEn falls in (1), the category of generative models section that will contribute to the library of potential candidates. As a side note, Lambo, a method we compared to, uses a BO inspired acquisition function to guide the search for better designs, and we do compare to it (favorably), but, we must highlight the difference, **PropEn and Lambo are generative models not BO/AL methods. We hope this clarifies the differences between the two frameworks and highlights their complementarity.

**[matching and diffusion models]**
Another remark raised by two reviewers gEZM and zYK4.
We found it surprising that the reviewers find a connection between PropEn and diffusion models, as these are very different frameworks. We do not see how matching can be straightforwardly incorporated into the training procedure of a diffusion model, where samples are successively noised until they resemble a simple base (usually Gaussian) distribution, followed by a reversed process that learns to denoise to the original state. Substituting the base distribution by a data distribution corresponds to a complete reformulation of the denoising diffusion framework and requires dealing with the lack of an easy-to-compute density.


We appreciate the reviewers insightful comments and suggestions, which have significantly improved our manuscript. We have carefully addressed each point in our responses and conducted additional experiments and analyses that we hope will make you reconsider and view our submission ready for acceptance. If there are any further questions, we are more than willing to address them.

---

### Decision · Program_Chairs · 2024-09-25

**Decision:**

Accept (poster)

**Comment:**

The paper introduces PropEn, a generative framework for design optimization that enhances properties of interest without requiring a discriminator, typically found in adversarial models. PropEn works by training on a "matched" dataset, where each example is paired with an improved version from its vicinity, and the generative model learns to approximate the gradient of the property function. The framework is claimed to be particularly effective in data-scarce scenarios and is validated through applications in airfoil design and protein optimization, with the latter showing superior performance in wetlab experiments.

The reviews highlight PropEn's simplicity and solid theoretical foundation as key strengths, particularly praising its applicability to real-world problems like protein optimization. However, they point out several weaknesses, including a lack of detailed implementation information, insufficient baselines in the airfoil task, and limited statistical analysis of the results. The rebuttal has provided a satisfying answer to the major concerns and I will recommend acceptance and ask the authors to incorporate the feedback.